# Notch signaling and Bsh homeodomain activity are integrated to diversify *Drosophila* lamina neuron types

**Chundi Xu[1]\*, Tyler B Ramos[1], Owen J Marshall[2], Chris Q Doe[1]\***

[1]Institute of Neuroscience, Howard Hughes Medical Institute, University of Oregon, Eugene, United States; [2]Menzies Institute for Medical Research, University of Tasmania, Hobart, Australia

**Abstract** Notch signaling is an evolutionarily conserved pathway for specifying binary neuronal fates, yet how it specifies different fates in different contexts remains elusive. In our accompanying paper, using the *Drosophila* lamina neuron types (L1-L5) as a model, we show that the primary homeodomain transcription factor (HDTF) Bsh activates secondary HDTFs Ap (L4) and Pdm3 (L5) and specifies L4/L5 neuronal fates. Here we test the hypothesis that Notch signaling enables Bsh to differentially specify L4 and L5 fates. We show asymmetric Notch signaling between newborn L4 and L5 neurons, but they are not siblings; rather, Notch signaling in L4 is due to Delta expression in adjacent L1 neurons. While Notch signaling and Bsh expression are mutually independent, Notch is necessary and sufficient for Bsh to specify L4 fate over L5. The Notch^ON L4, compared to Notch^OFF L5, has a distinct open chromatin landscape which allows Bsh to bind distinct genomic loci, leading to L4-specific identity gene transcription. We propose a novel model in which Notch signaling is integrated with the primary HDTF activity to diversify neuron types by directly or indirectly generating a distinct open chromatin landscape that constrains the pool of genes that a primary HDTF can activate.

**\*For correspondence:**
cxu3@uoregon.edu (CX);
cdoe@uoregon.edu (CQD)

**Competing interest:** The authors declare that no competing interests exist.

## eLife assessment

This paper explores how Notch activity acts together with homeodomain transcription Bsh factors to establish distinct cell fates (L4 vs L5) in the visual system of *Drosophila*. The findings are **important** and have theoretical or practical implications beyond a single subfield. The methods, data, and analyses are **compelling** and support the claims with only minor weaknesses.

## Introduction

The extraordinary computational power of our brain depends on the vast diversity of neuron types characterized initially by transcription factor (TF) combinatorial codes, followed by neuron-type-specific functional attributes such as cell surface molecules, neurotransmitters, and ion channels. It has been well documented how initial neuronal diversity is generated: in both invertebrate and vertebrate, spatial and temporal factors act combinatorially in progenitors to generate diverse and molecularly distinct progeny, and asymmetric Notch signaling between two newborn sister neurons further diversifies neuron types (***Bayraktar and Doe, 2013***; ***Bello-Rojas and Bagnall, 2022***; ***Doe, 2017***; ***Erclik et al., 2017***; ***Holguera and Desplan, 2018***; ***Peng et al., 2007***; ***Pierfelice et al., 2011***; ***Sen et al., 2019***; ***Spana and Doe, 1996***). Yet, it remains unclear how Notch signaling controls binary neuronal fate. Notch signaling is an evolutionarily conserved pathway that controls many aspects of nervous system development, and the outcome of Notch signaling often depends on its context-dependent

integration with other pathways (*Bray, 2016*; *Louvi and Artavanis-Tsakonas, 2006*). While most progenitor spatial and temporal factors are not maintained in neurons, our companion paper shows that the primary HDTF Bsh (Brain-specific homeobox) is persistently expressed in *Drosophila* lamina neurons, allowing it to couple initial fate decision to subsequent circuit formation and expression of a spectrum of neuronal functional genes, such as neurotransmitters and ion channels (*Xu et al., 2023*). This leads to the hypothesis that Notch signaling integrates with the primary HDTF activity to diversify neuron types. To test this, we use the *Drosophila* lamina, the first ganglion in the optic lobe, to ask whether and how Notch signaling acts with the Bsh HDTF to diversify lamina neuron types.

The *Drosophila* lamina has only five intrinsic neuron types (L1-L5), which are analogous to bipolar cells in the vertebrate visual system (*Sanes and Zipursky, 2010*). During late larval and early pupal stages, lamina progenitor cells (LPCs) give rise to L1-L5 neurons (*Fernandes et al., 2017*; *Huang et al., 1998*). The cell bodies of each lamina neuron type are localized in a layer-specific manner. L2/L3 cell bodies are intermingled in the most distal layer while L1, L4, and L5 form distinct layers progressively more proximal (*Tan et al., 2015*; *Xu et al., 2023*). Each lamina neuron type expresses unique TF markers: L1, L2, and L3 neurons express Zfh1/Svp, Bab2, and Zfh1/Erm, respectively (*Tan et al., 2015*; *Xu et al., 2023*), while L4 and L5 neurons express the HDTFs Bsh/Ap and Bsh/Pdm3, respectively (*Hasegawa et al., 2013*; *Tan et al., 2015*; *Xu et al., 2023*). Our accompanying paper shows that Bsh is initiated in LPCs and maintained in L4 and L5 neurons, while Ap and Pdm3 are initiated in L4 and L5 neurons, respectively (*Xu et al., 2023*). Based on their initiation order, we refer to Bsh as a 'primary' HDTF and Ap/Pdm3 as 'secondary' HDTFs (*Xu et al., 2023*). Bsh activates Ap and Pdm3 and specifies L4 and L5 fates (*Xu et al., 2023*). However, it remains unknown how a single primary HDTF Bsh activates two different secondary HDTFs and specifies two distinct neuron types (L4 and L5). Could Notch signaling distinguish L4 and L5 fates?

Here, we elucidate the role of Notch and the primary HDTF Bsh in specifying L4 and L5 neuronal fates, resulting in the following findings. (1) Differential Notch signaling distinguishes L4 and L5 neurons. Unlike in the medulla, central brain, and ventral nerve cord (*Lee et al., 2020*; *Li et al., 2013*; *Mark et al., 2021*), this is not due to an asymmetric partition of a Notch pathway component between sister neurons. Newborn L4 and L5 neurons are differentially exposed to the Notch ligand Delta expressed in L1, leading to asymmetric Notch activation in L4 but not L5. (2) Previously, the relationship between the primary HDTF Bsh and Notch signaling was unknown. Here, we show while Notch and Bsh expression are mutually independent, they act together to differentially specify L4 and L5 fates. (3) How Notch controls binary neuronal fates has remained elusive. We hypothesize that Notch signaling might regulate chromatin accessibility. Indeed, we find that compared to Notch$^{OFF}$ L5 neurons, Notch$^{ON}$ L4 neurons exhibit a distinct open chromatin landscape which shapes distinct Bsh genome-binding loci, resulting in L4-specific gene transcription. We propose that Notch signaling regulates the chromatin landscape in L4 neurons, leading to L4-distinct Bsh genomic binding and L4-specific Bsh-dependent gene expression.

## Results

### Notch signaling is activated in newborn L4 neurons but not L5

Binding of a Notch ligand in one cell to Notch in an adjacent cell results in the translocation of the Notch intracellular domain (N-ICD) into the nucleus and the subsequent transcription of Notch target genes, including Hey (*Monastirioti et al., 2010*; *Figure 1A*). To test whether Notch signaling is activated in newborn lamina neurons, we stained for Hey as a reporter of active Notch signaling. We found that Hey is colocalized with Bsh$^+$ L4 newborn neurons but not Bsh$^+$ L5 neurons (*Figure 1B–B" and D*). We also examined the expression of another Notch target gene, *E(spl)-mγ* (*Almeida and Bray, 2005*), and found that it was not expressed in lamina neurons (*Figure 1—figure supplement 1*). The primary HDTF Bsh activates the secondary HDTFs Ap and Pdm3 in L4 and L5, respectively (*Xu et al., 2023*). Here, we show that Notch signaling is activated prior to the initiation of these secondary HDTFs, suggesting a potential causal relationship between asymmetric Notch signaling and differential secondary HDTF activation in L4 and L5 neurons (*Figure 1C–D*). Together, we conclude that there is differential Notch activity, with newborn L4, but not L5, expressing the Notch reporter Hey.

In most regions of the central nervous system, ganglion mother cells (GMCs) undergo asymmetric terminal divisions to make Notch$^{ON}$/Notch$^{OFF}$ sibling neurons with distinct cell fates (*Lee et al., 2020*;

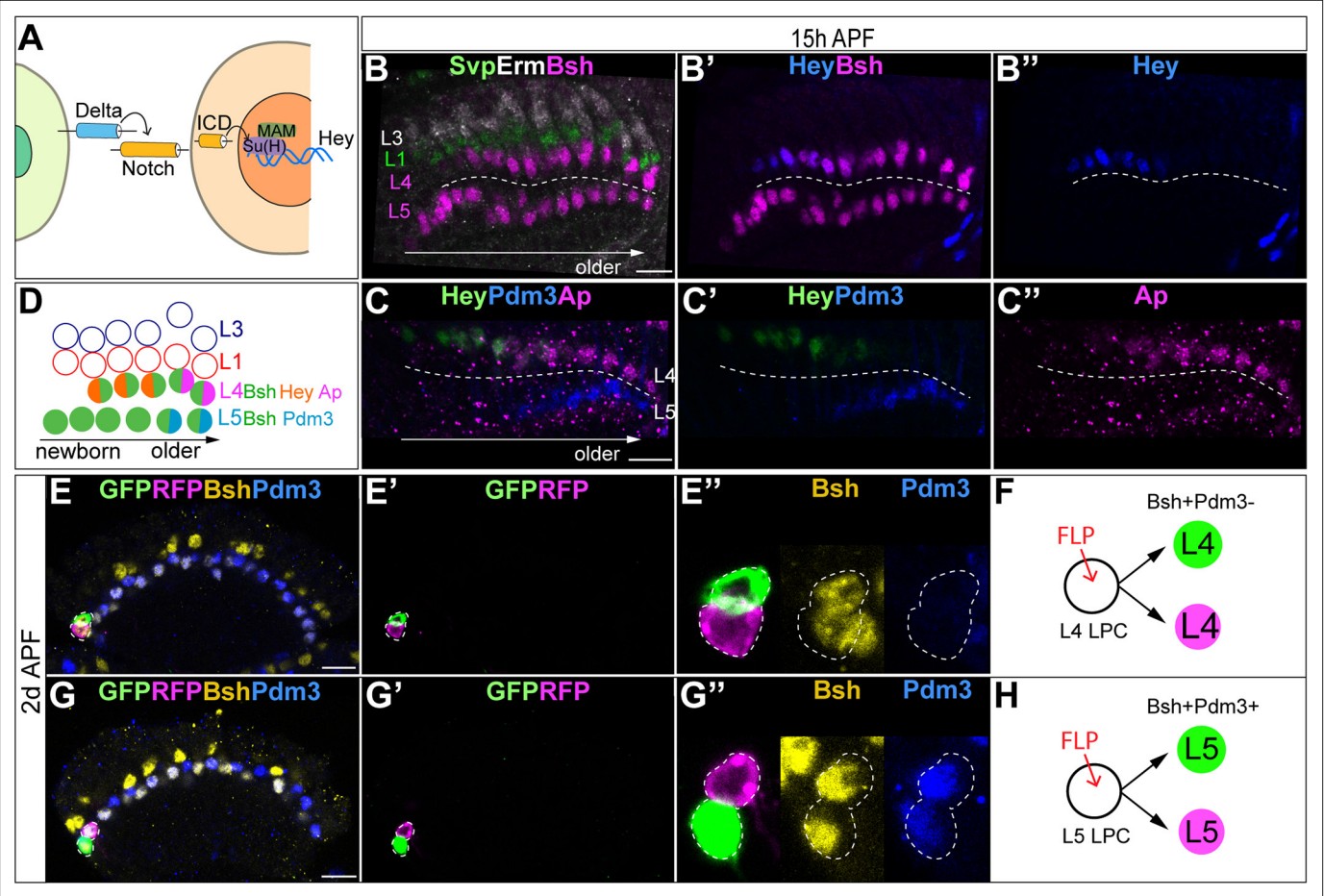

**Figure 1.** Notch signaling is activated in newborn L4 neurons but not L5. (**A**) Schematic of Notch signaling pathway. (**B-B"**) Hey as a reporter of active Notch signaling is only expressed in newborn L4 neurons but no other lamina neurons at 15 hr APF. Here and below, scale bar: 10 µm, n≥5 brains. Dashed line delineates the boundary between Bsh+ L4 and Bsh+ L5 neurons. (**C, D**) Hey is expressed prior to the activation of the secondary HDTFs Ap and Pdm3 at 15 hr APF. The dashed line delineates the boundary between Bsh+ L4 and Bsh+ L5 neurons. (**E–H**) Using twin-spot MARCM, two sibling neurons generated by one progenitor are traced. RFP and GFP cells are either both L4 neurons (Bsh+Pdm3-) or both L5 neurons (Bsh+Pdm3+). N=4. The dashed line outlines RFP+ and GFP+ cell bodies.

The online version of this article includes the following figure supplement(s) for figure 1:

**Figure supplement 1.** E(spl)-mγ is not expressed in lamina neurons.

Li et al., 2013; Mark et al., 2021). To test whether L4 and L5 are Notch[ON]/Notch[OFF] siblings, we used twin-spot MARCM (*Yu et al., 2009*) to trace two sibling neurons, which are divided from a single LPC cell, as a GFP and RFP pair. If L4 and L5 neurons are siblings generated from an LPC asymmetric division, we predict L4 and L5 to be an invariant RFP and GFP pair. In contrast, we found the GFP and RFP pair are either both L4 neurons or both L5 neurons (*Figure 1E–H*). This rules out an obligate asymmetric division to produce L4/L5 siblings and suggests that differential Hey expression is due to differential contact with an extrinsic Notch ligand.

## L1 neurons express Delta and activate Notch signaling in adjacent L4 neurons

We tested the possibility that the asymmetric Notch signaling between L4 and L5 is due to an extrinsic exposure of L4 to Delta expression, one of the two Notch ligands in *Drosophila* (*Muskavitch, 1994*). We found that during lamina neurogenesis, Delta is specifically expressed in L1 neurons which are adjacent to L4 neurons but not L5 (*Figure 2A–B*). Furthermore, we found that Delta is also expressed in the Tailless[+] (Tll[+]) LPCs in the same row as the L1 neurons, suggesting that LPCs may be more heterogeneous than previously thought (*Apitz and Salecker, 2014*; *Huang et al.,*

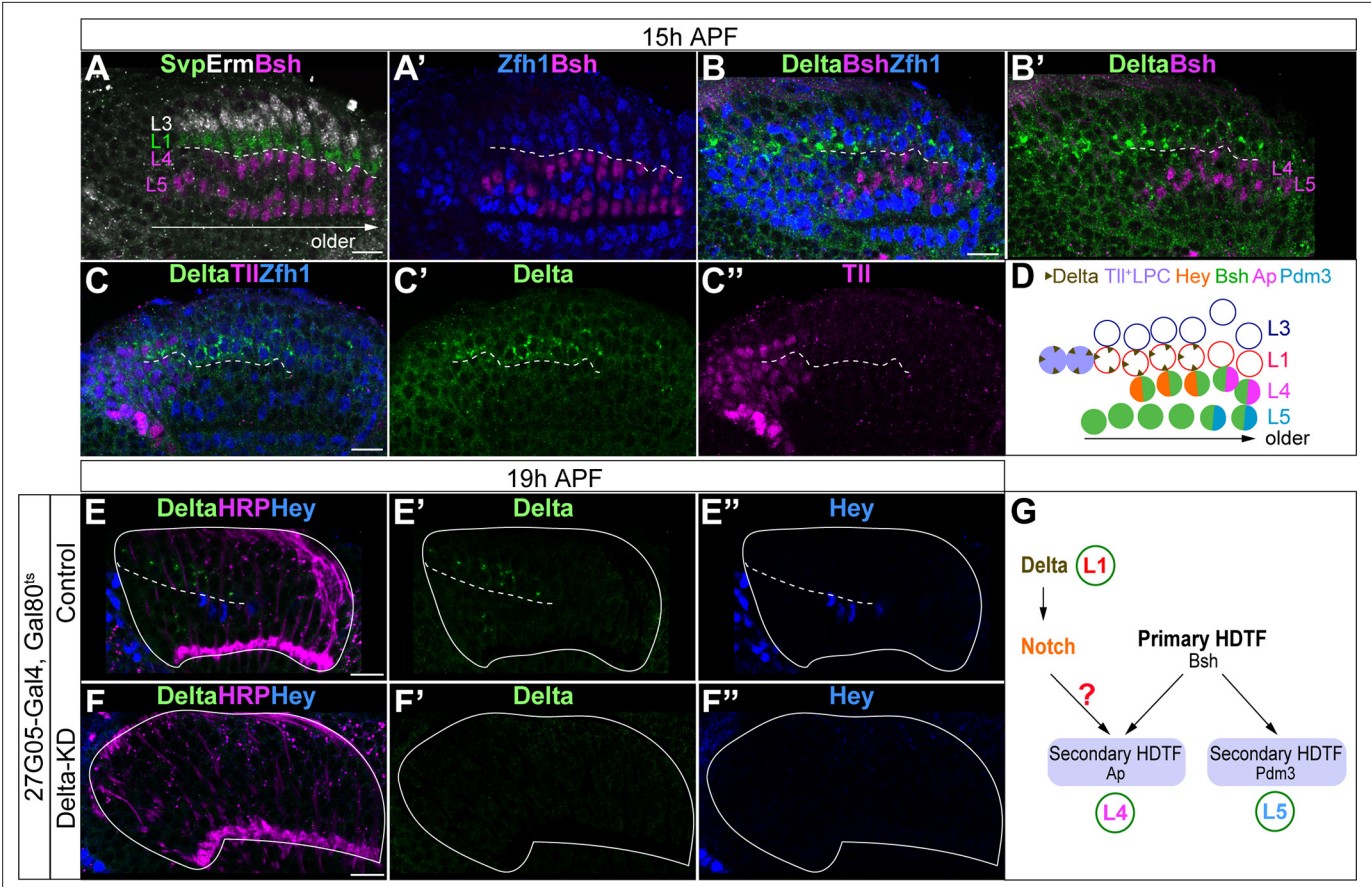

**Figure 2.** L1 neurons express Delta and activate Notch signaling in adjacent L4 neurons. (**A, A'**) Svp+Zfh1+L1 neurons are adjacent to both newborn L3 and L4 neurons. Newborn L3 neurons (Erm+) are localized strictly above (distal to) L1 neurons (Svp+Zfh1+), and L1 neurons are localized strictly above (distal to) L4 neurons. Here and below, scale bar: 10 µm, n≥5 brains. The dashed line delineates the boundary between L1 (Svp+Zfh1+) and L4 (Bsh+) cell bodies. (**B, B'**) Delta is expressed in Zfh1+ L1 neurons which are adjacent to Bsh+ L4 neurons. The dashed line delineates the boundary between L1 (Zfh1+) and L4 (Bsh+) cell bodies. (**C-C''**) Delta is also expressed in a subset of LPCs (Tll+). The dashed line highlights Delta+ cell bodies. (**D**) Summary of A-C data; triangles represent Delta expression. (**E-F''**) Delta-KD (27G05-Gal4, tubP-GAL80[ts], UAS-Delta-RNAi) results in loss of Delta and Hey expression in lamina. HRP labels the axons of the photoreceptors, which represent the lamina column. A solid white line outlines the lamina and a dashed line delineates the boundary between Delta+ cells and Hey+ cells. (**G**) Summary.

The online version of this article includes the following figure supplement(s) for figure 2:

**Figure supplement 1.** Svp is not required to initiate or maintain Delta expression in L1 neurons.

*1998*; *Figure 2C–D*). To test whether the L1-specific transcription factor Svp is required to activate or maintain Delta expression in L1, we knocked down Svp expression in the lamina. We found that Svp expression is indeed almost gone following Svp-knockdown (Svp-KD), yet Delta and Hey expression remains unaffected (*Figure 2—figure supplement 1*), suggesting that Delta expression in L1 neurons might be inherited from Delta-expressing LPCs instead of being activated by Svp. Together, we found that L1 neurons and a subset of LPCs express Delta, and Delta expression in L1 neurons does not depend on Svp.

Next, we asked whether Delta expression in L1 neurons is required to activate Notch signaling in adjacent newborn L4 neurons. We knocked down Delta expression in lamina using the LPC driver 27G05-Gal4 and Delta-RNAi. Indeed, Delta-KD shows the absence of Delta expression in the lamina (*Figure 2E' and F'*). Importantly, reduced Delta expression abolished Hey expression in the newborn L4, showing that Delta expression in L1 neurons is required for activating Notch in L4 (*Figure 2E–F''*). Taken together, we conclude that Delta expressed in L1 neurons activates Notch signaling in L4 but not L5 neurons, making the Delta-Notch pathway a strong candidate for acting together with Bsh activity to differentially specify L4 and L5 fates (*Figure 2G*).

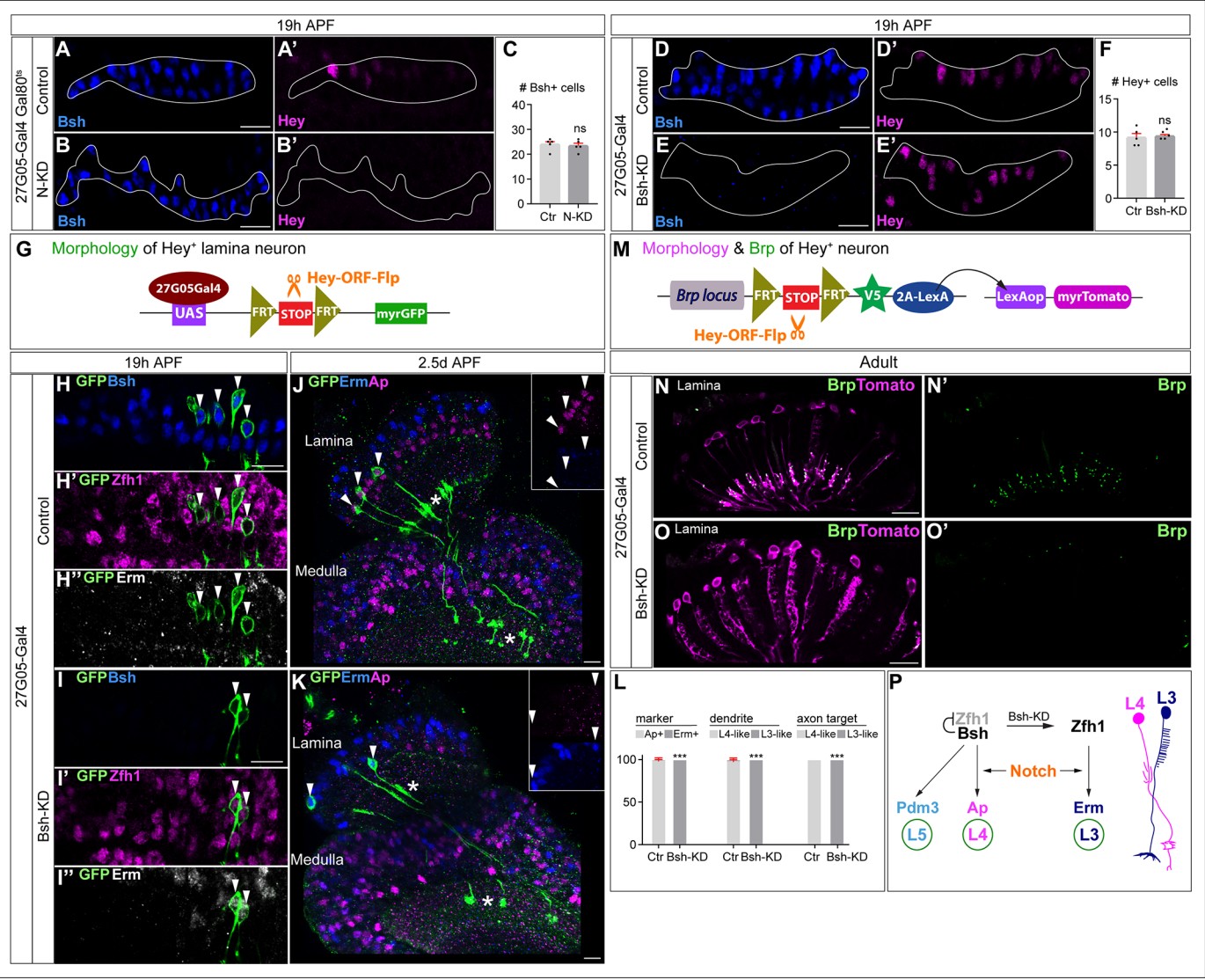

**Figure 5.** Notch activation and Bsh expression are mutually independent; Notch signaling without Bsh specifies L3 neuron type. (**A–F**) Notch activation and Bsh expression are mutually independent. (**A–C**) N-KD in lamina (27G05-Gal4, tubP-GAL80[ts], UAS-Notch-RNAi) results in loss of Hey expression without affecting Bsh expression. (**C**) Quantification (single optical section). Here and below, scale bar, 10 μm, n≥5 brains. (**D–F**) Bsh-KD in lamina (27G05-Gal4>UAS-Bsh-RNAi) results in loss of Bsh expression without affecting Hey expression. (**F**) Quantification (single optical section). (**G–L**) Notch signaling without Bsh specifies L3 neuron type. (**G**) Schematic of the genetics used to trace the morphology of Hey+ lamina neurons. (**H, I**) In controls, GFP+ neurons express the L4 marker Bsh (white arrowhead) at 19 hr APF. In Bsh-KD, Bsh expression becomes absent in lamina and GFP+ cells now express L3 markers Zfh1 and Erm (white arrowhead). (**J–L**) In controls, GFP + cells express the L4 marker Ap (white arrowhead) and have L4 morphology (asterisk) at 2.5d APF. In Bsh-KD, GFP+ cells express L3 marker Erm (white arrowhead) and adopt L3 morphology (asterisk). (**L**) Quantification for (**J**) and (**K**) (single optical section). (**M–O**) (**M**) Schematic of the genetics used to trace the morphology and presynaptic sites of Hey+ neurons. (**N, O**) In controls, Tomato+ neurons have L4 morphology and presynaptic sites (Brp) in the lamina. In Bsh-KD, Tomato+ neurons adopt L3 morphology and connectivity, which lacks presynaptic localization in the lamina (*Xu et al., 2019*). (**P**) Summary. Data are presented as mean ± SEM. Each dot represents each brain. n=5 brains in (**C**), (**F**), and (**L**). ***p<0.001, ns = not significant, unpaired t-test.

## Bsh without Notch signaling activates Pdm3 and specifies L5 neuronal fate

To test whether the asymmetric Notch signaling between newborn L4 and L5 enables Bsh to differentially specify distinct L4 and L5 fates, we first performed Notch loss of function in the lamina using 27G05-Gal4 and UAS-Notch-RNAi. We found that this genotype caused embryonic lethality, so to preserve Notch function during early development, we performed conditional Notch-KD (N-KD) using Gal80[ts]. We inactivated the Gal80[ts] activity, which abrogates the suppression of 27G05-Gal4, from the

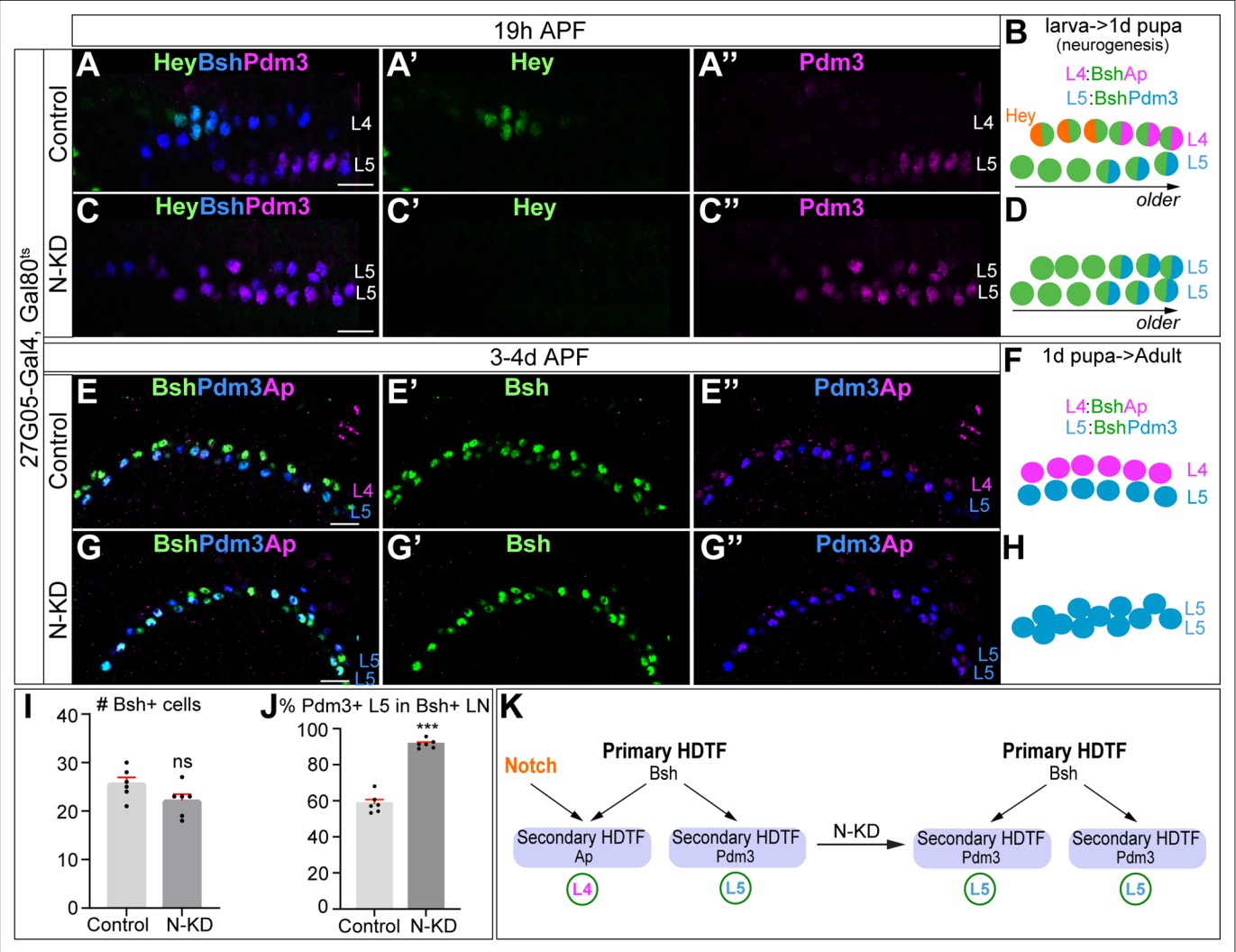

**Figure 3.** Bsh without Notch signaling activates Pdm3 and specifies L5 neuronal fate. (**A–D**) N-KD in lamina (27G05-Gal4, tubP-GAL80[ts], UAS-Notch-RNAi) shows Hey expression is absent, and Bsh only activates Pdm3 and specifies L5 neuronal fate during lamina neurogenesis (19 hr APF). Here and below, scale bar, 10 μm, n≥5 brains. (**E–J**) N-KD in lamina shows Bsh+ lamina neurons are mainly Pdm3+ L5 neurons, though the number of Bsh+ lamina neurons remains unaffected at 3-4d APF. (**I–J**) Quantification (single optical section). (**K**) Summary Data are presented as mean ± SEM. Each dot represents each brain. n=6 brains in (**I**), (**J**). ***p<0.001, ns = not significant, unpaired t-test.

The online version of this article includes the following figure supplement(s) for figure 3:

**Figure supplement 1.** 27G05-Gal4 is inserted in between two Notch target genes; Bsh without Notch signaling activates Pdm3 and specifies L5 morphology.

**Figure supplement 2.** Bsh specifies L5 neuronal fate over L4 following Delta-KD.

beginning of the third instar stage to ensure the N-KD during lamina neurogenesis. During lamina neurogenesis, in controls, the Notch reporter Hey is expressed in newborn L4 neurons, followed by Bsh activation of Ap and Pdm3 and specification of L4 and L5 fates, respectively (*Xu et al., 2023*; *Figure 3A–B*, *Figure 3—figure supplement 1B–B"*). In contrast, in N-KD, Hey expression becomes absent, and Bsh now activates Pdm3 but not Ap and specifies L5 fate (*Figure 3C–D*, *Figure 3—figure supplement 1B–C"*). During late pupal stages after the completion of lamina neurogenesis, control Bsh+ lamina neurons include both Ap+ L4 neurons and Pdm3+ L5 neurons (*Figure 3E–F*). In N-KD, however, Bsh+ lamina neurons are mainly Pdm3+ L5 neurons, although the number of Bsh+ lamina neurons remains unaffected, suggesting that more L5 neurons are generated at the expense of L4 in N-KD (*Figure 3E–J*). Interestingly, Bsh+ cell bodies in N-KD settle together to mix in a single layer, in contrast to Bsh+ cell bodies in controls segregating into two distinct layers with L4 distal to L5

(*Figure 3E–H*). This suggests that L4 and L5 cell bodies in controls might express different cell surface molecules to achieve their neuron-type-specific settling layer.

Similarly, Delta-KD results in more L5 neurons generated at the expense of L4 (*Figure 3—figure supplement 2*). Note that ectopic Ap expression in L5 is caused by the 27G05-Gal4 line alone (*Figure 3E–G"*, *Figure 3—figure supplement 2A–B"*), probably due to its genome insertion site in the Notch target gene *E(spl)m4-BFM* (*Figure 3—figure supplement 1A*), but this does not affect our conclusion that Bsh without Notch signaling activates Pdm3 and specifies L5 fate. Taken together, we conclude that in the absence of Notch signaling, the primary HDTF Bsh activates the secondary HDTF Pdm3 and specifies L5 neuronal fate (*Figure 3K*).

## Bsh with Notch signaling activates Ap and specifies L4 neuronal fate

To test whether Notch signaling is sufficient for Bsh to activate Ap and specify L4 neuronal fate, we first performed Notch gain of function broadly in the LPCs and lamina neurons. We used 27G05-Gal4 and Gal80$^{ts}$ to express the Notch intracellular domain (N-ICD; a constitutively active form of Notch) from the middle of lamina neurogenesis at 0 hr APF (*Figure 4—figure supplement 1A*). Following Notch misexpression, we observed ectopic Hey expression in newborn L5 neurons generated after 0 hr APF, showing that Notch signaling was active in L5 neurons (*Figure 4—figure supplement 1B–C"*). Furthermore, Notch misexpression results in ectopic Bsh$^+$ Pdm3$^-$ L4 neurons and a loss of Bsh$^+$ Pdm3$^+$ L5 neurons (*Figure 4—figure supplement 1D–G*). Thus, broad activation of Notch signaling in lamina is sufficient for Bsh to activate Ap and specify L4 neuronal fate at the expense of L5 neuronal fate.

Next, we wanted to activate Notch signaling precisely in newborn L5 neurons using Bsh-Gal4. We confirmed that Bsh-Gal4 is turned on in newborn L5 and prior to Pdm3 initiation, making it the perfect driver for Notch gain of function in newborn L5 (*Figure 4—figure supplement 2A–B"*). In controls, newborn L4 neurons express Hey followed by Ap, whereas newborn L5 neurons lack Hey expression and subsequently express Pdm3 (*Xu et al., 2023*; *Figure 4A–B*, *Figure 4—figure supplement 2C–C"*). In contrast, expression of N-ICD in newborn 'L5 neurons' resulted in ectopic expression of Hey followed by expression of the L4 marker Ap (*Figure 4C–D*, *Figure 4—figure supplement 1D-D"*). During late pupal stages after the completion of lamina neurogenesis, control Bsh$^+$ lamina neurons include both Ap$^+$ L4 neurons and Pdm3$^+$ L5 neurons (*Figure 4E–F*). In N-ICD, however, Bsh$^+$ lamina neurons are mainly Ap$^+$ L4 neurons, though the number of Bsh$^+$ lamina neurons remains unaffected, suggesting that more L4 neurons are generated at the expense of L5 in N-ICD (*Figure 4E–J*). These data support our conclusion that Notch signaling in newborn neurons is sufficient for Bsh to specify L4 neuronal fate at the expense of L5 neuronal fate (*Figure 4K*).

To test whether Bsh with Notch signaling specifies L4 morphology and connectivity, we took advantage of a Hey-ORF-Flp transgene (*Mark et al., 2021*) to trace the morphology of Hey$^+$ neurons with myristoylated Tomato and the Hey$^+$ neuron presynaptic sites with Bruchpilot (Brp) using the STaR method (*Chen et al., 2014*; *Xu et al., 2019*; *Figure 4—figure supplement 3A*). L4 neurons are the only lamina neurons that express Hey (*Figure 1B'*). Indeed, in controls, Tomato + neurons, which trace Hey$^+$ neurons, exhibit L4-specific morphology and connectivity in lamina: neurites and presynaptic Brp in the proximal lamina (*Figure 4—figure supplement 3B–C*). Following N-ICD misexpression in newborn L5, Tomato$^+$ neurons, which trace the endogenous and ectopic Hey$^+$ neurons, have L4-like morphology and connectivity: neurites and presynaptic Brp in the proximal lamina (*Figure 4—figure supplement 3D–E*). The slight deviations from control L4 morphology following N-ICD misexpression in newborn L5 might be due to a higher Notch signaling level in N-ICD misexpression compared to control. Taken together, we conclude that in the presence of Notch signaling, the primary HDTF Bsh activates the secondary HDTF Ap and specifies L4 TF marker, morphology, and connectivity (*Figure 4K*).

## Notch signaling and Bsh expression are mutually independent

Notch signaling is widely used in newborn neurons to specify binary neuronal fates, and HDTFs are broadly expressed in neurons (*Hobert, 2021*; *Jukam and Desplan, 2010*; *Louvi and Artavanis-Tsakonas, 2006*; *Spana and Doe, 1996*), yet the relationship between Notch signaling and HDTF expression has remained elusive. Here, we find that the secondary HDTFs Ap and Pdm3 are activated in a Notch-dependent manner: Notch signaling upregulates Ap expression and downregulates Pdm3 expression (*Figures 3 and 4*). To test the relationship between Notch signaling and

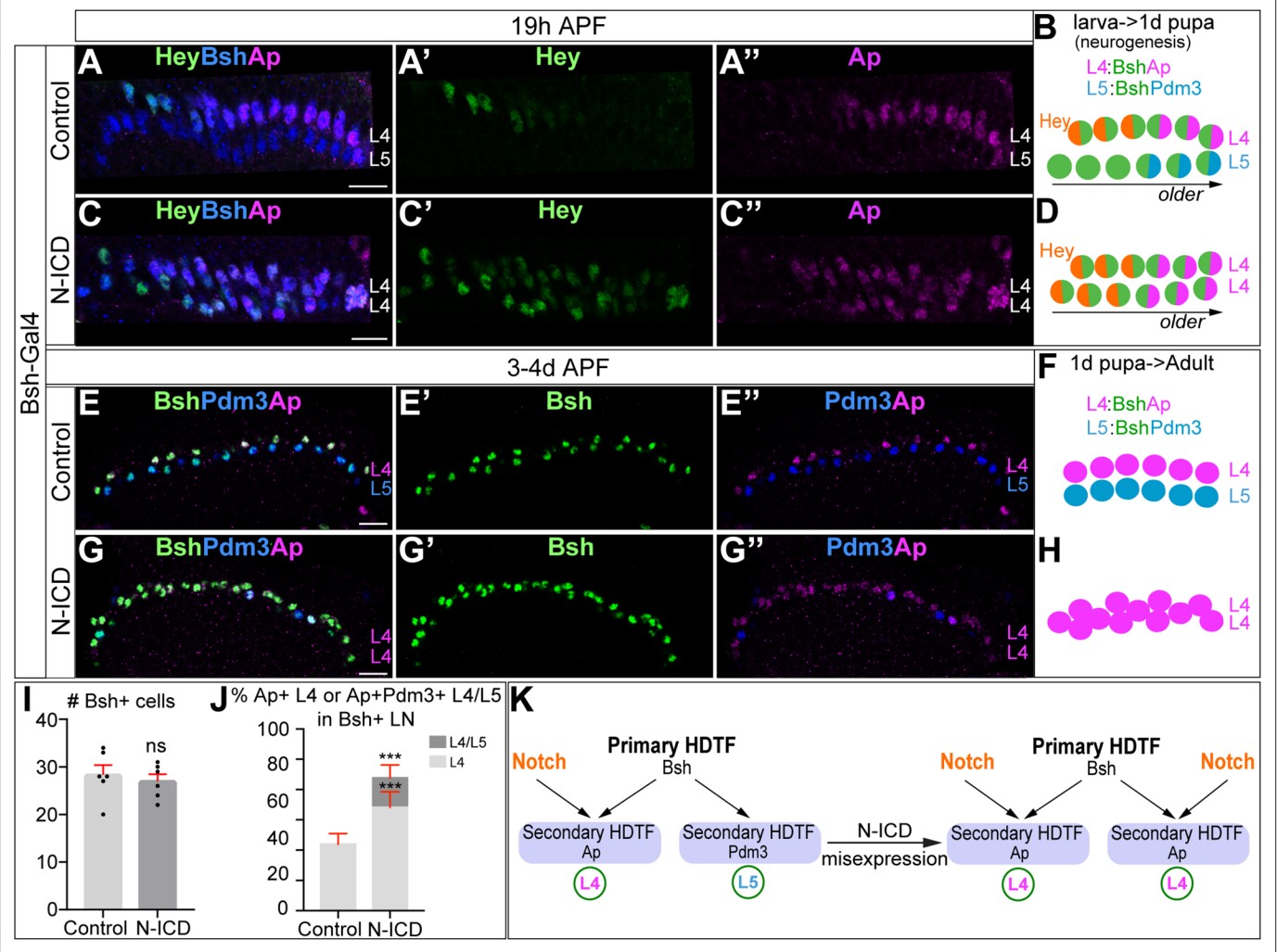

**Figure 4.** Bsh with Notch signaling activates Ap and specifies L4 neuronal fate. (**A–D**) Ectopic expression of N-ICD in newborn L5 neurons (Bsh-Gal4 >UAS-N-ICD) results in ectopic Hey and Ap activation and an increased number of L4 neurons at 19 hr APF. Here and below, scale bar, 10 μm, n≥5 brains. (**E–J**) N-ICD shows Bsh+ lamina neurons are mainly Ap+ L4 neurons, though the number of Bsh+ lamina neurons remains unaffected at 3-4d APF. (**I–J**) Quantification (single optical section). (**K**) Summary. Data are presented as mean ± SEM. Each dot represents each brain. n=6 brains in (**I**), (**J**). ***p<0.001, ns = not significant, unpaired t-test.

The online version of this article includes the following figure supplement(s) for figure 4:

**Figure supplement 1.** Temporally restricted activation of Notch signaling by 27G05-Gal4 enables Bsh to specify L4 neuronal fate over L5.

**Figure supplement 2.** Bsh-Gal4 is expressed in newborn L5 neurons; Bsh does not activate L5 marker Pdm3 when Notch signaling is activated in newborn L5 by Bsh-Gal4 and UAS-N-ICD.

**Figure supplement 3.** Bsh-Gal4 activation of Notch signaling in newborn L5 neurons specifies L4-like morphology and presynaptic sites.

the primary HDTF Bsh expression, we performed conditional N-KD during lamina neurogenesis. This knockdown was highly effective, with Hey becoming undetectable, yet the number of Bsh+ cells remains unaffected (*Figure 5A–C*). Thus, Bsh expression is independent of Notch signaling. Conversely, we used 27G05-Gal4 and Bsh-RNAi to knock down Bsh in LPCs. This resulted in Bsh becoming undetectable, yet Hey expression was unaffected (*Figure 5D–F*). Thus, Notch signaling and Bsh expression are independent. Taken together, we conclude that while the expression of the secondary HDTFs depends on Notch signaling, the expression of primary HDTF Bsh and Notch signaling are mutually independent.

## Notch signaling without Bsh specifies L3 neuron type

Our observation that Notch signaling remains unaffected following the loss of Bsh raises an interesting question: what is the identity of the Hey[+] neurons in the absence of Bsh? To address this question, we used genetic methods (schematic in *Figure 5G*) to trace Hey[+] lamina neurons with myristoylated GFP. L4 neurons are the only lamina neurons expressing Hey. Indeed, in controls, GFP + cells, which trace Hey[+] neurons, express the L4 marker Bsh but not L3 markers Zfh1 or Erm (*Figure 5H–H''*). In Bsh-KD, Bsh expression is lost, and GFP[+] cells instead express the L3 markers Zfh1 and Erm (*Figure 5I–I''*). Zfh1 is required for L1 and L3 fates, and Bsh suppresses Zfh1 to inhibit ectopic L1 and L3 fates (*Xu et al., 2023*). Together, we conclude that in the absence of Bsh, Notch signaling with ectopic Zfh1 specifies L3 neuronal fate.

We next tested whether Hey[+] neurons in Bsh-KD adopt L3 morphology. In controls, GFP[+] neurons express the L4 marker Ap and exhibit L4-specific morphology; they project neurites into the proximal lamina, and their axons target two layers in medulla (*Figure 5J*). In Bsh-KD, GFP[+] neurons express the L3 marker Erm and adopt L3-specific morphology: one-direction (comb-like) neurites in lamina and axon targeting to a single layer in medulla (*Xu et al., 2019*; *Figure 5K and L*). We conclude that in the absence of Bsh, Notch signaling with ectopic Zfh1 specifies L3 neuronal fate and L3-specific morphology.

To test whether Hey[+] neurons in Bsh-KD adopt L3 connectivity, we examined the location of presynaptic sites of Hey[+] neurons. We took advantage of Hey-ORF-Flp (*Mark et al., 2021*) to label the presynaptic marker Brp of Hey[+] neurons with the STaR method (*Chen et al., 2014*; *Xu et al., 2019*) (schematic in *Figure 5M*). In controls, Tomato[+] neurons in lamina exhibit L4-specific morphology and connectivity, with neurites and presynaptic Brp in the proximal lamina (*Figure 5N*). In contrast, following Bsh-KD, the Tomato[+] neurons adopt L3-specific morphology and connectivity: comb-like neurites in lamina and no significant Brp in lamina (*Xu et al., 2019*; *Figure 5O*). Taken together, we conclude that in the absence of Bsh, Notch signaling with ectopic Zfh1 specifies L3 TF marker, morphology, and connectivity (*Figure 5P*).

## The Notch[ON] L4 has correlated open chromatin, Bsh-bound loci, and transcription profile that is distinct from the Notch[OFF] L5

Bsh specifies two distinct neuron types: Notch[ON] L4 and Notch[OFF] L5. How are Notch signaling and Bsh activity integrated to specify L4 neuron type over L5? Bsh binds numerous L4 feature genes: ion channels, synaptic organizers, cytoskeleton regulators, synaptic recognition molecules, neuropeptide/receptor, and neurotransmitter/receptor (*Xu et al., 2023*). This raises the hypothesis that Notch signaling might generate an L4-distinct open chromatin landscape which determines the accessibility of Bsh target genes, resulting in L4-specific Bsh-dependent gene transcription. To determine if L4 and L5 have differing open chromatin regions and Bsh genome-binding sites, we used Targeted DamID (TaDa) (*Marshall et al., 2016*; *Southall et al., 2013*) to profile both open chromatin (Dam-alone binding; *Aughey et al., 2018*) and Bsh genome-binding loci (Dam:Bsh binding; *Figure 6A*). We performed these experiments with precise spatial and temporal control: only in L4 neurons (using L4-Gal4), which we reported in our accompanying paper (*Xu et al., 2023*) or L5 neurons (using L5-Gal4; *Luo et al., 2020*; *Nern et al., 2008*) at the time of synapse formation (46–76 hr APF; *Figure 6—figure supplement 1A–B''*). We confirmed the function of the Bsh:Dam fusion protein previously (*Xu et al., 2023*). Here for L5 neurons, we performed three biological replicates, which had high reproducibility (*Figure 6—figure supplement 1C*).

Our data show that the Notch[ON] L4 neurons, compared to the Notch[OFF] L5, have distinct open chromatin regions, which is consistent with the known role of Notch signaling driving histone acetylation, a marker of active enhancers, in cultured *Drosophila* and mammalian cells (*Oswald et al., 2001*; *Skalska et al., 2015*; *Wang et al., 2014*; *Figure 6B*, *Supplementary file 1*). Indeed, we found the canonical binding motif of Suppressor of Hairless (Su(H)), the DNA-binding partner of Notch (*Bray and Furriols, 2001*), is more likely to be enriched in L4-distinct open chromatin regions (*Figure 6—figure supplement 2A*). To test whether L4-distinct open chromatin regions result in L4-specific gene transcription, we took advantage of recently published lamina neuron single-cell RNA sequencing (scRNA-seq) data from the same developmental stage (GEO: GSE190714; *Jain et al., 2022*). Indeed, our data show the significant enrichment of L4-specific transcribed genes in L4-distinct open chromatin regions (*Figure 6B*, *Supplementary file 2*). Similarly, we found that L5-specific transcribed genes are enriched

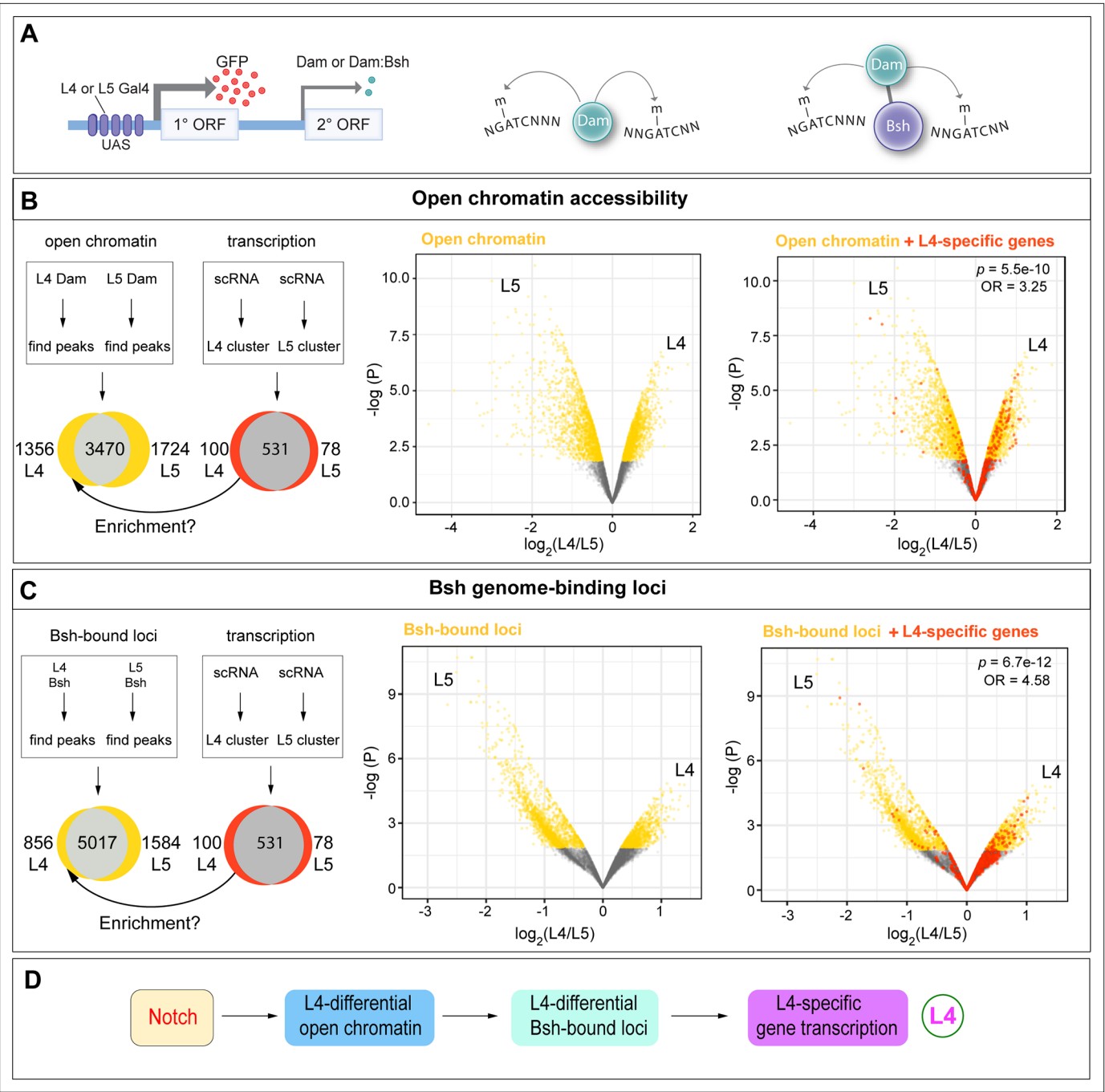

**Figure 6.** The Notch[ON] L4 has correlated open chromatin, Bsh-bound loci, and transcription profile that is distinct from the Notch[OFF] L5. (**A**) Schematic of the Targeted DamID method. Upon GAL4 induction, low levels of either Dam or Bsh:Dam are expressed, allowing genome-wide open chromatin and Bsh binding targets to be identified. (**B**) L4 and L5 neurons show distinct open chromatin regions (yellow), and L4-specific transcribed genes (red) are enriched in L4-distinct open chromatin regions. (**C**) L4 and L5 neurons show distinct Bsh-bound loci (yellow), and L4-specific transcribed genes (red) are enriched in L4-distinct Bsh-bound loci. p Values from Fisher's exact test; odds ratios (OR) expressed as L4/L5 in all cases. (**D**) Model.

The online version of this article includes the following figure supplement(s) for figure 6:

**Figure supplement 1.** Six to 60 Gal4 used for TaDa experiment is specifically expressed in L5 neurons at 46–76 hr APF.

**Figure supplement 2.** L4-distinct open chromatin regions are more likely to contain the Su(H) binding motif; L5-specific transcribed genes are correlated with L5-distinct open chromatin regions but not L5-distinct Bsh-bound loci.

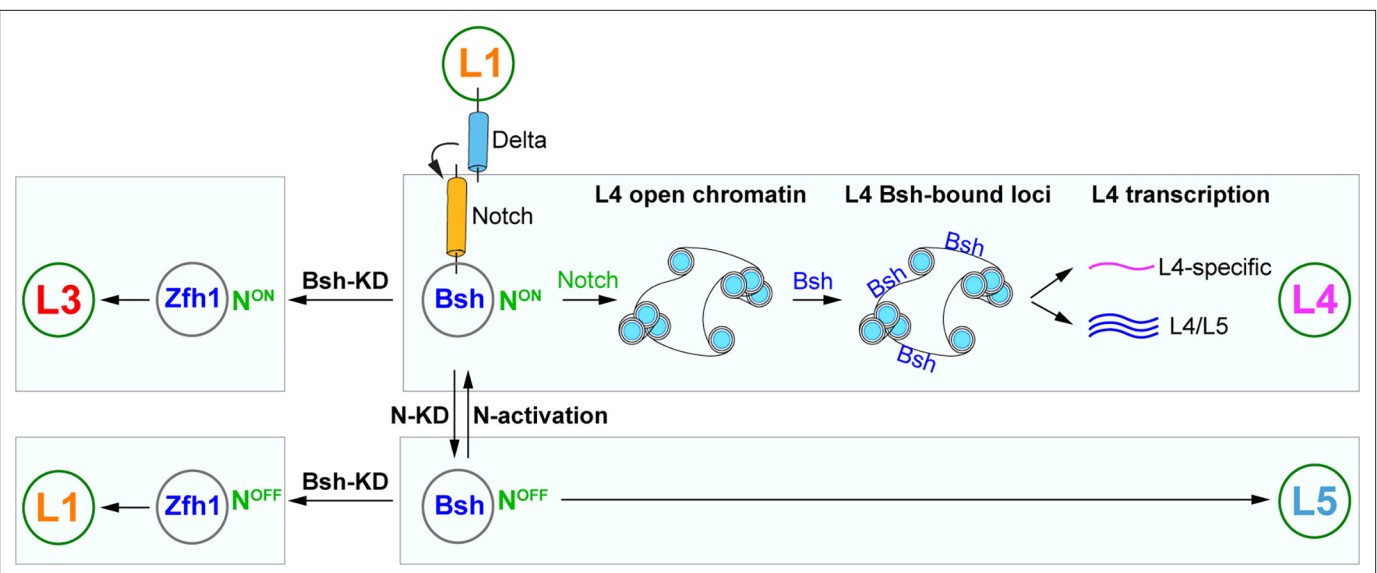

**Figure 7.** Model; see the discussion section for details.

in L5-distinct open chromatin regions (*Figure 6—figure supplement 2B*, *Supplementary file 3*). Together, our data suggest that Notch signaling directly or indirectly generates an L4-distinct open chromatin landscape which correlates with L4-specific gene transcription (*Figure 6D*).

Next, we tested the hypothesis that L4-distinct open chromatin landscape leads to L4-distinct Bsh genome-binding loci, which in turn results in L4-specific gene transcription. We found that L4 and L5 neurons indeed show distinct Bsh genome-binding loci (*Figure 6C*, *Supplementary file 4*). Furthermore, L4-distinct Bsh-bound loci, similar to L4-distinct open chromatin, are enriched for L4-specific transcribed genes (*Figure 6C*, *Supplementary file 2*). This suggests that L4-distinct open chromatin, via L4-distinct Bsh-bound loci, leads to L4-specific gene transcription. Interestingly, we did not see the significant enrichment of L5-specific transcribed genes in L5-distinct Bsh-bound loci, suggesting another unknown factor, for example the secondary HDTF Pdm3, might be required for L5-specific gene transcription (*Figure 6—figure supplement 2C*, *Supplementary file 3*). Taken together, we propose that Notch signaling creates a distinct open chromatin landscape leading to distinct Bsh-bound loci in L4 neurons, and thus generating L4-specific Bsh-dependent gene transcription (*Figure 6D*).

## Discussion

Notch signaling is an evolutionarily conserved pathway for controlling binary cell fates in worms, flies, and mammals (*Artavanis-Tsakonas et al., 1999*; *Bray, 2016*; *Ehebauer et al., 2006*; *Jukam and Desplan, 2010*; *Louvi and Artavanis-Tsakonas, 2006*; *Pierfelice et al., 2011*; *Pinto-Teixeira et al., 2018*), yet its mechanism has been unclear. Here, we discovered that transient Notch signaling in newborn neurons diversifies neuron types by integrating with the primary HDTF. The primary HDTF Bsh with Notch signaling specifies L4 fate, while Bsh alone specifies L5 fate. The Notch[ON] L4, compared to Notch[OFF] L5, has a distinct open chromatin landscape which allows Bsh to bind distinct genomic loci, leading to L4-specific identity gene transcription. Notch signaling and Bsh expression are mutually independent. In the absence of Bsh, the Notch[ON] and Notch[OFF] neurons with ectopic HDTF Zfh1 adopt L3 and L1 (*Xu et al., 2023*) fates, respectively (*Figure 7*). Together, based on our findings, we propose a novel model in which the unique combination of the primary HDTF and open chromatin landscape specifies distinct neuron types.

Does Notch signaling regulate chromatin landscape independently of a primary HDTF? It seems likely based on the following observation: in the *Drosophila* larval ventral nerve cord, motor and sensory hemilineages are generated in a Notch-dependent manner; all premotor neurons are Notch[ON], with Notch[ON] neurons sharing similar anatomical and functional properties, despite expressing distinct HDTFs (*Doe, 2017*; *Heckscher et al., 2014*; *Mark et al., 2021*), possibly due to the common

chromatin landscape regulated by Notch signaling. An important future direction would be to test whether Notch[ON] neurons retain the same chromatin landscape in the absence of the primary HDTF, for example Bsh.

Despite the important roles of Notch signaling in development, its function in lamina development was unknown. Here, we discovered that L1 neurons express Delta and activate Notch signaling in adjacent newborn L4 but not in the more distant L5. L3 is also adjacent to the Delta[+] L1; might L3 be a Notch[ON] neuron? If so, it would use a different effector, as Hey is not expressed in newborn L3 neurons. Yet, it seems likely for several reasons: (1) like L4, newborn L3 neurons are strictly adjacent to Delta[+] L1; (2) Asymmetric Notch signaling between L3 and L1 would allow Zfh1 to differentially specify L3 and L1 neuronal fates; (3) in Bsh-KD, Notch signaling with ectopic Zfh1 specifies ectopic L3 neuron type. To tackle this, an intriguing approach would be profiling the genome-binding targets of endogenous Notch in newborn neurons. This will identify novel genes as Notch signaling reporters in neurons. Besides lamina neurons, we also detected Delta expression in a small pool of LPCs in the L1 row and the Notch target E(spl)-mγ expression in the L4 row, raising the hypothesis that Delta-expressing LPCs may rise to L1 neurons while E(spl)-mγ-expressing LPCs may rise to L4 neurons. This also suggests that LPCs are more heterogeneous than previously thought (*Apitz and Salecker, 2014*; *Fernandes et al., 2017*). It would be interesting to characterize the molecular heterogeneity of LPCs using scRNA-seq and explore the potential role of Notch signaling in this population.

It is well known that Notch signaling controls binary cell fates. Yet, its mechanism remains unclear. We found that Notch signaling and primary HDTF Bsh activity are integrated to specify two distinct neuron types: L4 and L5. Notch signaling directly or indirectly generates an L4-distinct open chromatin landscape which specifies L4-distinct Bsh-bound loci, resulting in L4-specific Bsh-dependent gene transcription. This is consistent with in vitro findings of Notch function. In cultured mammalian cells, upon Notch activation, Notch transcription complexes recruit p300, which acetylates H3K27 and produce a large increase in H3K27 acetylation levels, an active enhancer marker, across the entire breadth of super-enhancers (*Oswald et al., 2001*; *Wang et al., 2014*). In cultured *Drosophila* cells, Notch activation induces a robust increase in H3K56 acetylation (*Bray, 2016*; *Skalska et al., 2015*). One intriguing future direction would be testing whether Notch signaling directly generates an open chromatin landscape in vivo.

Our brain function depends on the vast diversity of neuron types. The role of TFs in specifying neuron types has been well-studied. For example, each temporal transcription factor (TTF) acts transiently in progenitors to generate specific neuron types during its expression window (*Doe, 2017*; *Isshiki et al., 2001*; *Li et al., 2013*). While most TTFs are not maintained in neurons, their function is likely maintained by another TF, for example HDTF, which is persistently expressed in neurons. Indeed, HDTFs function as terminal selectors and control the expression of neuronal identity genes (*Cros and Hobert, 2022*; *Hobert, 2021*; *Howell et al., 2015*; *Reilly et al., 2022*; *Reilly et al., 2020*). The primary HDTF Bsh specifies L4 and L5 neuronal fates (*Xu et al., 2023*). However, the role of open chromatin in specifying neuron type is less well characterized. In vitro, during the programming of mouse embryonic stem cells to neurons, proneural factors Ascl1 and Neurogenin2 establish distinct chromatin landscapes, resulting in induced neurons with different subtype identities (*Aydin et al., 2019*; *Wapinski et al., 2013*). Here, we found that Bsh with Notch signaling specifies L4 fate, while Bsh alone specifies L5 fate. The Notch[ON] L4, compared to Notch[OFF] L5, has a distinct open chromatin landscape which allows Bsh to bind distinct genomic loci, leading to L4-specific identity gene transcription. Furthermore, in Bsh-KD, L3 and L1 are ectopically generated at the expense of L4 and L5, respectively (*Figure 7*). This elegant one-to-one fate switch (L4 >L3; L5 >L1) raises an open question: would newborn L4 and L3 share the open chromatin landscape, which is distinct from the one shared by newborn L5 and L1? Our findings provide a testable model that the unique combination of the primary HDTF and open chromatin landscape specifies distinct neuron types.

## Data availability

All resources will be provided upon request. DamID data in this publication have been deposited in NCBI's GEO and are accessible through GEO Series accession number GSE247239. All original code has been deposited on GitHub (https://github.com/marshall-lab/Xu_et_al_2023, copy archived at *Marshall lab, 2023*).

# Materials and methods

## Key resources table

| Reagent type (species) or resource | Designation | Source or reference | Identifiers | Additional information |
|---|---|---|---|---|
| Strain, strain background (*D. melanogaster*) | 10xUAS-IVS-myristoylated-GFP | Bloomington *Drosophila* Stock Center | RRID: BDSC_32199 | w[1118]; P{y[+t7.7] w[+mC]=10XUAS-IVS-myr::GFP}su(Hw)attP5 |
| Strain, strain background (*D. melanogaster*) | *R27G05GAL4* | Bloomington *Drosophila* Stock Center | RRID: BDSC_48073 | w[1118]; P{y[+t7.7] w[+mC]=GMR27 G05-GAL4}attP2 |
| Strain, strain background (*D. melanogaster*) | UAS-Bsh-RNAi | Bloomington *Drosophila* Stock Center | RRID: BDSC_29336 | y[1] v[1]; P{y[+t7.7] v[+t1.8]=TRiP.JF02498}attP2 |
| Strain, strain background (*D. melanogaster*) | yw; UAS-mCD8-GFP, UAS-rCD2i, FRT40A/CyO; TM3, Sb/TM6B, Tb | Bloomington *Drosophila* Stock Center | RRID: BDSC_56185 | y[1] w[*]; P{y[+t7.7] w[+mC]=UAS-mCD8.GFP.UAS-rCD2i}attP40 P{ry[+t7.2]=neoFRT}40 A/CyO; TM3, Sb[1]/TM6B, Tb[1] |
| Strain, strain background (*D. melanogaster*) | yw, hsFlp122; frt40a, UAS-CD2-RFP, UAS-GFP-miRNA | Gift from Claude Desplan Lab | | yw, hsFLP; uas-CD2::RFP, UAS-GFP-miRNA/cyo; tm2/tm6b,tb |
| Strain, strain background (*D. melanogaster*) | UAS-Notch-RNAi | Bloomington *Drosophila* Stock Center | RRID: BDSC_33611 | y[1] v[1]; P{y[+t7.7] v[+t1.8]=TRiP.HMS00001}attP2 |
| Strain, strain background (*D. melanogaster*) | UAS-Notch-ICD | *Struhl and Greenwald, 2001* | | ;UAS-Notch-ICD; +/+ |
| Strain, strain background (*D. melanogaster*) | Hey-ORF-T2A-FlpD5 | *Mark et al., 2021* | | w; Hey-ORF-T2A-FLP(RFP+)/cyo, wg-LacZ; +/+ |
| Strain, strain background (*D. melanogaster*) | UAS-FRT-STOP-FRT-myrGFP | Bloomington *Drosophila* Stock Center | RRID: BDSC_55810 | w[1118]; P{y[+t7.7] w[+mC]=10XUAS(FRT.stop)GFP.Myr}su(Hw)attP5 |
| Strain, strain background (*D. melanogaster*) | w; Brp-FRT-STOP-FRT-smGdP-v5-T2A-LexA; LexAop-Tomato | *Peng et al., 2018*; *Xu et al., 2019* | | w; Brp-FRT-STOP-FRT-smGdP-v5-T2A-LexA/cyo; LexAop-myrtdTomato/tm6b |
| Strain, strain background (*D. melanogaster*) | UAS-Svp-RNAi | Bloomington *Drosophila* Stock Center | RRID: BDSC_28689 | y[1] v[1]; P{y[+t7.7] v[+t1.8]=TRiP.JF03105}attP2 |
| Strain, strain background (*D. melanogaster*) | *6–60* Gal4 | Gift from Lawrence Zipursky | | W; Bl/cyo; 6–60 Gal4/tm6b |
| Strain, strain background (*D. melanogaster*) | Bsh-L-Gal4 | Gift from Makoto Sato | | ;Bsh-L-Gal4/cyo; +/+ |
| Strain, strain background (*D. melanogaster*) | tubP-GAL80[ts] | Bloomington *Drosophila* Stock Center | RRID: BDSC_7017 | w[*]; P{w[+mC]=tubP-GAL80[ts]}2/TM2 |
| Strain, strain background (*D. melanogaster*) | UAS-Zfh1-RNAi | Vienna *Drosophila* Resource Center | VDRC 103205 | P{KK109931}VIE-260B |
| Strain, strain background (*D. melanogaster*) | 31C06-Gal4, UAS-myristoylated-tdTomato | Gift from Lawrence Zipursky | | ;Bl/cyo; 31c06-Gal4, UAS- myristoylated-tdTomato/tm6b |
| Strain, strain background (*D. melanogaster*) | E(spl)-mγ-GFP | Gift from Sarah Bray (*Almeida and Bray, 2005*) | | w; +/+; E(spl)mγ-GFP/tm6b |
| Strain, strain background (*D. melanogaster*) | VALIUM20-mCherry | Bloomington *Drosophila* Stock Center | RRID: BDSC_35785 | y[1] sc[*] v[1] sev[21]; P{y[+t7.7] v[+t1.8]=VALIUM20-mCherry}attP2 |

*Continued on next page*

*Continued*

| Reagent type (species) or resource | Designation | Source or reference | Identifiers | Additional information |
|---|---|---|---|---|
| Strain, strain background (*D. melanogaster*) | Bsh-ORF-3XHA (86Fb) | FlyORF Webshop | Cat#F000054 | M{UAS-bsh.ORF.3xHA.GW}ZH-86Fb |
| Strain, strain background (*D. melanogaster*) | flyORF-TaDa | Bloomington *Drosophila* Stock Center | RRID: BDSC_91637 | w[1118]; M{RFP[3xP3.PB] w[+mC]=FlyORF-TaDa}ZH-86Fb |
| Strain, strain background (*D. melanogaster*) | hs-FlpD5; FlyORF-TaDa | Bloomington *Drosophila* Stock Center | RRID: BDSC_91638 | w[1118]; P{y[+t7.7] w[+mC]=hs-FLPD5}attP40; M{RFP[3xP3.PB] w[+mC]=FlyORF-TaDa}ZH-86Fb |
| Strain, strain background (*D. melanogaster*) | Bsh-TaDa | *Xu et al., 2023* | | w; +/CyO; UAS-GFP-Bsh-DAM/tm6b |
| Antibody | Chicken polyclonal | Abcam | Cat#ab13970, RRID_300798 | Anti-GFP (1:1000) |
| Antibody | Rabbit polyclonal | Medical & Biological Laboratories Co. | Code#PM005 | Anti-RFP (1:500) |
| Antibody | Rabbit polyclonal | Gift from Claude Desplan (*Özel et al., 2021*) | | Anti-Bsh (1:1000) |
| Antibody | Guinea pig polyclonal | Gift from Lawrence Zipursky (*Tan et al., 2015*) & Makoto Soto | | Anti-Bsh (1:1000) |
| Antibody | Rat monoclonal | This study | | Anti-Bsh (1:750); see methods section- generating Bsh antibody |
| Antibody | Rabbit polyclonal | Gift from Markus Affolter (*Bieli et al., 2015*) | | Anti-Apterous (1:1000) |
| Antibody | Rat monoclonal | Gift from Cheng-Ting Chien (*Chen et al., 2012*) | | Anti-Pdm3 (1:200) |
| Antibody | Rabbit polyclonal | Gift from Cheng-Yu Lee (*Janssens et al., 2014*) | | Anti-Erm (1:100) |
| Antibody | Rat monoclonal | Gift from Jing Peng (*Santiago et al., 2021*) | | Anti-Erm (1:70) |
| Antibody | Mouse monoclonal | Developmental Studies Hybridoma Bank | Cat#Seven-up D2D3, RRID_2618079 | Anti-Svp (1:10) |
| Antibody | Rabbit polyclonal | Gift from James Skeath (*Tian et al., 2004*) | | Anti-Zfh1 (1:1000) |
| Antibody | Rabbit polyclonal | Asian Distribution Center for Segmentation Antibodies | Code#812 | Anti-Tailless (1:200) |
| Antibody | Mouse monoclonal | Developmental Studies Hybridoma Bank | Cat#Elav-9F8A9, RRID: AB_528217 | Anti-Elav (1:200) |
| Antibody | Mouse monoclonal | Bio-Rad Laboratories | Cat#MCA1360A647, RRID: AB_770156 | Anti-V5-TAG:Alexa Fluor 647 (1:300) |
| Antibody | Mouse monoclonal | Developmental Studies Hybridoma Bank | Cat#nc-82, RRID: AB_2314866 | Anti-Brp (1:50) |
| Antibody | Mouse monoclonal | Developmental Studies Hybridoma Bank | Cat#C594.9B, RRID: AB_528194 | Anti-Delta (1:10) |
| Antibody | Mouse monoclonal | TaKaRa | Cat#632543 | Anti-Cherry (1:500) |
| Antibody | Donkey polyclonal | Jackson ImmunoResearch Lab | Cat# 711-475-152 AB_2340616 | DyLight 405 anti-rabbit (1:400) |
| Antibody | Donkey polyclonal | Jackson ImmunoResearch Lab | Cat#703-545-155, RRID: AB_2340375 | Alexa Fluor 488 anti-chicken (1:400) |
| Antibody | Donkey polyclonal | Jackson ImmunoResearch Lab | Cat#706-545-148, RRID: AB_2340472 | Alexa Fluor 488 anti-guinea pig (1:400) |

*Continued on next page*

*Continued*

| Reagent type (species) or resource | Designation | Source or reference | Identifiers | Additional information |
|---|---|---|---|---|
| Antibody | Donkey polyclonal | Jackson ImmunoResearch Lab | Cat#711-545-152, RRID: AB_2313584 | Alexa Fluor 488 anti-rabbit (1:400) |
| Antibody | Donkey polyclonal | Jackson ImmunoResearch Lab | Cat#715-545-150, RRID: AB_2340846 | Alexa Fluor 488 anti-mouse (1:400) |
| Antibody | Donkey polyclonal | Jackson ImmunoResearch Lab | Cat#715-295-151, RRID: AB_2340832 | Rhodamine Red-X anti-mouse (1:400) |
| Antibody | Donkey polyclonal | Jackson ImmunoResearch Lab | Cat#712-295-153, RRID: AB_2340676 | Rhodamine Red-X anti-rat (1:400) |
| Antibody | Donkey polyclonal | Jackson ImmunoResearch Lab | Cat#711-295-152, RRID: AB_2340613 | Rhodamine Red-X anti-rabbit (1:400) |
| Antibody | Donkey polyclonal | Jackson ImmunoResearch Lab | Cat#706-295-148, RRID: AB_2340468 | Rhodamine Red-X donkey anti-guinea pig (1:400) |
| Antibody | Donkey polyclonal | Jackson ImmunoResearch Lab | Cat#711-605-152, RRID: AB_2492288 | Alexa Fluor 647 donkey anti-rabbit (1:400) |
| Antibody | Donkey polyclonal | Jackson ImmunoResearch Lab | Cat#715-605-151, RRID: AB_2340863 | Alexa Fluor 647 donkey anti-mouse (1:400) |
| Antibody | Donkey polyclonal | Jackson ImmunoResearch Lab | Cat#706-605-148, RRID: AB_2340476 | Alexa Fluor 647 anti-guinea pig (1:400) |
| Antibody | Donkey polyclonal | Jackson ImmunoResearch Lab | Cat#712-605-153, RRID: AB_2340694 | Alexa Fluor 647 anti-rat (1:400) |
| Sequence-based reagent | Oligonucloetide | Integrated DNA technologies | | DamID Adaptor (top strand): CTAATACGACTCACTATAGGGCAGCGTGGTCGCGGCCGAGGA |
| Sequence-based reagent | Oligonucloetide | Integrated DNA technologies | | DamID Adaptor (bottom strand): TCCTCGGCCG |
| Sequence-based reagent | Oligonucloetide | Integrated DNA technologies | | DamID Primer for PCR: GGTCGCGGCCGAGGATC |
| Commercial assay or kit | QIAamp DNA Micro Kit | Qiagen | Cat#56304 | |
| Commercial assay or kit | PCR Purification Kit | Qiagen | Cat#28104 | |
| Chemical compound, drug | EDTA | Sigma-Aldrich | Cat#E6758 | |
| Chemical compound, drug | DpnI and CutSmart buffer | NEB | Cat#R0176S | |
| Chemical compound, drug | DpnII and DpnII buffer | NEB | Cat#R0543S | |
| Chemical compound, drug | MyTaq HS DNA Polymerase | Bioline | Cat#BIO-21112 | |
| Chemical compound, drug | AlwI | NEB | Cat#R0513S | |
| Chemical compound, drug | RNase A (DNase free) | Roche | Cat#11119915001 | |
| Chemical compound, drug | T4 DNA ligase and 10 x buffer | NEB | Cat#M0202S | |
| Software, algorithm | Fiji | *Schindelin et al., 2012* | https://imagej.nih.gov/ij/download.html | |
| Software, algorithm | FastQC (v0.11.9) | The Babraham Bioinformatics group | https://www.bioinformatics.babraham.ac.uk/projects/download.html#fastqc | |

*Continued on next page*

*Continued*

| Reagent type (species) or resource | Designation | Source or reference | Identifiers | Additional information |
|---|---|---|---|---|
| Software, algorithm | damidseq_pipeline | *Marshall and Brand, 2015* | https://owenjm. github.io/damidseq_ pipeline/ | |
| Software, algorithm | Bowtie2 (v2.4.5) | *Langmead and Salzberg, 2012* | http://bowtie-bio. sourceforge.net/ bowtie2/index.shtml | |
| Software, algorithm | IGV (v.2.13.2) | *Robinson et al., 2011* | https://software. broadinstitute. org/software/igv/ download | |
| Software, algorithm | SAMtools (v1.15.1) | *Li et al., 2009* | http://www.htslib.org/ download/ | |
| Software, algorithm | deepTools (v3.5.1) | *Ramírez et al., 2016* | https://deeptools. readthedocs.io/en/ develop/content/ installation.html | |
| Software, algorithm | Find_peaks | *Marshall et al., 2016* | https://github.com/ owenjm/find_peaks; (*Marshall, 2016*) | |

## Contact for reagent and resource sharing

Further information and requests for resources and reagents should be directed to and will be fulfilled by the Lead Contact Chundi Xu (cxu3@uoregon.edu) or Chris Doe (cdoe@uoregon.edu).

## Experimental model and subject details

All flies were reared at 25 °C on standard cornmeal fly food, unless otherwise stated.

## Method details

### Animal collections

For all RNAi knockdown experiments without tubP-GAL80[ts], crosses were kept at 25 °C and the progeny were kept at 27.5 °C with 16:8 hours light-dark cycle from the embryo stage until dissection.

For the experiment R27G05GAL4>UAS-Notch-RNAi, tubP-GAL80[ts], the crosses were kept at 18 °C and the progeny were moved to 29.2 °C after 6 days in 18 °C. Pupae at 0 hr APF were staged at 29.2 °C and then dissected after 15 hr at 29.2 °C. Pupae 3-4d APF at 25 °C were dissected once they turned dark.

For the experiment 27G05GAL4>UAS-N-ICD, tubP-GAL80[ts], the crosses were kept at 18 °C and the progeny were moved to 29.2 °C at 0hAPF. Pupae at 19 hr APF or 3.75d APF were dissected.

For the experiment Bsh-Gal4 >UAS-N-ICD, the crosses were kept in 25 °C and the progeny were moved to 29.2 °C at early L3 stage for 15 hr (19 hr APF at 25 °C) or until dark (3-4d APF at 25 °C).

For twin-spot MARCM, the crosses (virgin: hsFlp122; frt40a, UAS-CD2-RFP, UAS-GFP-miRNA; Tm2/tm6b; male: w; frt40a, UAS-CD8-GFP, UAS-CD2-miRNA; 27g05Gal4/tm6b) were kept in 25 °C.

Progenies at 0 hr APF were heat shocked at 34 °C for 2 min and 50 s and dissected 24 hr after heat shock.

### Generating Bsh antibody

Bsh protein and antibody were made by GenScript (Piscataway, NJ) in guinea pigs, and immunized with the following portion of the Bsh open reading frame:

MHHHHHHAMLNEASLSPADAHAHANATTPTHSKAAAMASATTMLTTKTPFSIEHILFQNLNSAS
NNNNSSDTNGIAANTNNYAPKSSRNAVKSARSAFAHDNNPHKHPSQHSHPPQSHPPASASASAT
ATARSNQAASGYAGEDYGKSMHSTPRSNHHSRHGTSHYNGDQISQQLGSGAAQHPPVPTT
QPQPPPPPLNGGSGASNGVLYPNAPYTDHGFLQMTLGYLSPSSGTYKSVDPYFLSQASLFGGA
PFFGAPGCVPELALGLGMGVNALRHCRRRKARTVFSDPQLSGLEKRFEGQRYLSTPERVELATA

LGLSETQVKTWFQNRRMKHKKQLRRRDNANEPVDFSRSEPGKQPGEATSSSGDSKHGKLNPGSV
GGTPTQPTSEQQLQMCLMQQGYSTDDYSDLEADSGDEDNSSDVDIVGDAKLYQLT.

## Immunohistochemistry

Fly brains were dissected in Schneider's medium and fixed in 4% paraformaldehyde in phosphate buffered saline (PBS) for 25 min. After fixation, brains were quickly washed with PBS with 0.5% Triton X-100 (PBT) and incubated in PBT for at least 2 hr at room temperature. Next, samples were incubated in blocking buffer (10% normal donkey serum, 0.5% Triton X-100 in PBS) overnight at 4 °C. Brains were then incubated in primary antibody (diluted in blocking buffer) at 4 °C for at least two nights. Following primary antibody incubation, brains were washed with PBT. Next, brains were incubated in secondary antibody (diluted in blocking buffer) at 4 °C for at least 1 day. Following secondary antibody incubation, brains were washed with PBT and mounted in SlowFade Gold antifade reagent (Thermo Fisher Scientific, Waltham, MA). Images were acquired using a Zeiss 800 confocal and processed with Image J, Adobe Illustrator (San Jose, CA) and Adobe Photoshop (San Jose, CA).

## TaDa in L4 and L5 neurons at the time of synapse formation

For TaDa in L4, homozygous tubP-GAL80[ts]; 31C06-Gal4, UAS-myristoylated-tdTomato males were crossed to homozygous virgin females (FlyORF-TaDa line for Dam; Bsh-TaDa line for Bsh:Dam). For TaDa in L5, homozygous tubP-GAL80[ts]; 6–60 Gal4 UAS-myristoylated-tdTomato males were crossed to homozygous virgin females (FlyORF-TaDa line for Dam; Bsh-TaDa line for Bsh:Dam). Crosses were reared at 18 °C. To perform TaDa in L4 and L5 neurons during the synapse formation window, we collected pupae with the age of 46 hr APF and moved them to 29 °C to activate 31C06-Gal4 (L4) or 6–60 Gal4 (L5) for 24 hr (*Figure 6—figure supplement 1*). Then lamina were dissected (age equivalent at 25 °C: 76 hr APF) in cold PBS within one hour and stored at –20 °C immediately until sufficient laminae were collected for each group—about 70 lamina from 35 pupae. The published (*Marshall et al., 2016*) TaDa experimental pipeline was followed with a few modifications. Briefly, DNA was extracted using a QIAamp DNA Micro Kit (Qiagen, Germantown, MD) digested with DpnI (NEB, Ipswich, MA) overnight, and cleaned up using Qiagen PCR purification columns. DamID adaptors were ligated using T4 DNA ligase (NEB, Ipswich, MA) followed by DpnII (NEB, Ipswich, MA) digestion for 2 hr and PCR amplification using MyTaq HS DNA polymerase (Bioline, Memphis, Tennessee). The samples were sequenced on the NovaSeq at 118 base pairs and 27–33 million single-end reads per sample.

## Bioinformatic analysis

DamID-seq NGS sequencing reads were processed using damidseq_pipeline v1.5.3 (*Marshall and Brand, 2015*). For generating Bsh binding profile bedGraphs, the default settings were used. For generating Dam-alone coverage bedGraphs, the pipeline was run on the Dam-alone sequencing files with the `--just_coverage` setting. For both datasets (Bsh binding and Dam-alone), peaks were identified using find_peaks (*Marshall et al., 2016*) on separate replicates.

All downstream analyses were conducted using R (*Team RC, 2016*). Significant peaks from all replicates were combined using the GenomicRanges R package (*Lawrence et al., 2013*), and the average occupancy over each significant peak was determined for each replicate. As the overall Bsh binding profiles between L4 and L5 neurons were highly correlated, the binding profiles of all replicates were quantile normalized prior to further analysis.

Differentially open regions in the Dam-alone dataset, and differentially-bound regions in the Bsh binding dataset, were identified using NOIseq (*Tarazona et al., 2015*) as previously described (*Hatch et al., 2021*) at a threshold of q=0.85. Separately, peaks were assigned to genes, with all genes within 1 kb of a peak assigned as putative regulatory targets of that peak.

Tests for enrichment were conducted using Fisher's exact test, with contingency tables derived from differentially open or bound regions and the overlap of these with lineage-specific gene expression targets. The alternative hypothesis was that Bsh-bound targets or open chromatin would be enriched for genes expressed in the same lineage. Odds ratios were expressed as L4/L5 for all samples.

Motif searches were conducted using the core YGTGRGAAM motif, obtained as a PWM from CisBP (*Weirauch et al., 2014*) (Motif ID: M10415_2.00), using the R Biostrings package (*Pagès et al.,*

*2023*). For motif searches, open chromatin regions were limited to a maximum size of 1 kb around the center of the region. Motifs with a match score ≥ 85% were counted as a match.

## Statistical analysis

Statistics were performed using either Microsoft Excel or Prism (GraphPad, San Diego, California) software. Unpaired t-test was used, unless otherwise noted. Data are presented as mean ± SEM unless otherwise noted. A 95% confidence interval was used to define the level of significance. *$p<0.05$, **$p<0.01$, ***$p<0.001$, ns = not significant. All other relevant statistical information can be found in the figure legends.

## Acknowledgements

We thank Claude Desplan, Larry Zipursky, Makoto Sato, Markus Affolter, Jing Peng, Cheng-Yu Lee, James Skeath, Cheng-Ting Chien for antibodies; and Kristen Lee, Peter Newstein, Noah Dillon, Sen-Lin Lai for comments on the manuscript. Stocks obtained from the Bloomington *Drosophila* Stock Center were used in this study.

## Additional information

### Funding

| Funder | Grant reference number | Author |
| --- | --- | --- |
| Howard Hughes Medical Institute | | Chris Q Doe |
| National Health and Medical Research Council | Ideas Grant APP1185220 | Owen J Marshall |

The funders had no role in study design, data collection and interpretation, or the decision to submit the work for publication.

### Author contributions

Chundi Xu, Conceptualization, Resources, Data curation, Formal analysis, Supervision, Validation, Investigation, Visualization, Methodology, Writing – original draft, Writing – review and editing; Tyler B Ramos, Resources, Data curation, Formal analysis, Validation, Investigation, Writing – review and editing; Owen J Marshall, Resources, Data curation, Software, Formal analysis, Writing – review and editing; Chris Q Doe, Resources, Supervision, Funding acquisition, Visualization, Methodology, Project administration, Writing – review and editing

### Author ORCIDs

Chundi Xu ⬥ https://orcid.org/0000-0002-1056-8893
Chris Q Doe ⬥ https://orcid.org/0000-0001-5980-8029

Reviewer #1 (Public Review): https://doi.org/10.7554/eLife.90136.3.sa1
Reviewer #2 (Public Review): https://doi.org/10.7554/eLife.90136.3.sa2
Author Response https://doi.org/10.7554/eLife.90136.3.sa3

## Additional files

### Supplementary files

- Supplementary file 1. Comparison of open chromatin regions between L4 and L5 neurons.
- Supplementary file 2. Genes that are expressed in L4 neurons but not L5.
- Supplementary file 3. Genes that are expressed in L5 neurons but not L4.
- Supplementary file 4. Comparison of Bsh genome-binding loci between L4 and L5 neurons.
- MDAR checklist

## Data availability

DamID data in this publication have been deposited in NCBI's GEO and are accessible through GEO Series accession number GSE247239.

The following dataset was generated:

| Author(s) | Year | Dataset title | Dataset URL | Database and Identifier |
|---|---|---|---|---|
| Xu C, Ramos TB, Marshall O, Doe CQ | 2023 | Notch signaling and Bsh homeodomain activity are integrated to diversify *Drosophila* lamina neuron types | https://www.ncbi.nlm.nih.gov/geo/query/acc.cgi?acc=GSE247239 | NCBI Gene Expression Omnibus, GSE247239 |

The following previously published dataset was used:

| Author(s) | Year | Dataset title | Dataset URL | Database and Identifier |
|---|---|---|---|---|
| Jain S, Lin Y, Kurmangaliyev YZ, Zipursky SL | 2021 | A global timing mechanism regulates cell-type specific wiring programs | https://www.ncbi.nlm.nih.gov/geo/query/acc.cgi?acc=GSE190714 | NCBI Gene Expression Omnibus, GSE190714 |

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
