## [Editor Report · eLife assessment]

This paper explores how Notch activity acts together with homeodomain transcription Bsh factors to establish distinct cell fates (L4 vs L5) in the visual system of *Drosophila*. The findings are **important** and have theoretical or practical implications beyond a single subfield. The methods, data, and analyses are **compelling** and support the claims with only minor weaknesses.

---

## [Referee Report · Reviewer #1 (Public Review)]

Like the "preceding" co-submitted paper, this is again a very strong and interesting paper in which the authors address a question that is raised by the finding in their co-submitted paper - how does one factor induce two different fates. The authors provide an extremely satisfying answer - only one subset of the cells neighbors a source of signaling cells that trigger that subset to adopt a specific fate. The signal here is Delta and the read-out is Notch, whose intracellular domain, in conjunction with, presumably, SuH cooperates with Bsh to distinguish L4 from L5 fate (L5 is not neighbored by signal-providing cells). Like the back-to-back paper, the data is rigorous, well-presented and presents important conclusions. There's a wealth of data on the different functions of Notch (with and without Bsh). All very satisfying.

I have again one suggestion that the authors may want to consider discussing. I'm wondering whether the open chromatin that the author convincingly measure is the CAUSE or the CONSEQUENCE of Bsh being able to activate L4 target genes. What I mean by this is that currently the authors seem to be focused on a somewhat sequential model where Notch signaling opens chromatin and this then enables Bsh to activate a specific set of target genes. But isn't it equally possible that the combined activity of Bsh/Notch(intra)/SuH opens chromatin? That's not a semantic/minor difference, it's a fundamentally different mechanism, I would think. This mechanism also solves the conundrum of specificity - how does Notch know which genes to "open" up? It would seem more intuitive to me to think that it's working together with Bsh to open up chromatin, with chromatin accessibility than being a "mere" secondary consequence. If I'm not overlooking something fundamental here, there is actually also a way to distinguish between these models - test chromatin accessibility in a Bsh mutant. If the author's model is true, chromatin accessibility should be unchanged.

I again finish by commending the authors for this terrific piece of work.

---

## [Referee Report · Reviewer #2 (Public Review)]

Summary:

In this work, the authors explore how Notch activity acts together with Bsh homeodomain transcription factors to establish L4 and L5 fates in the lamina of the visual system of *Drosophila*. They propose a model in which differential Notch activity generates different chromatin landscapes in presumptive L4 and L5, allowing the differential binding of the primary homeodomain TF Bsh (as described in the co-submitted paper), which in turn activate downstream genes specific to either neuronal type. The requirement of Notch for L4 vs. L5 fate is well supported, and complete transformation from one cell type into the other is observed when altering Notch activity. However, the role of Notch in creating differential chromatin landscapes is not directly demonstrated. It is only based on correlation, but it remains a plausible and intriguing hypothesis.

Strengths:

The authors are successful in characterizing the role of Notch to distinguish between L4 and L5 cell fates. They show that the Notch pathway is active in L4 but not in L5. They identify L1, the neuron adjacent to L4 as expressing the Delta ligand, therefore being the potential source for Notch activation in L4. Moreover, the manuscript shows molecular and morphological/connectivity transformations from one cell type into the other when Notch activity is manipulated.

Using DamID, the authors characterize the chromatin landscape of L4 and L5 neurons. They show that Bsh occupies distinct loci in each cell type. This support their model that Bsh acts as a primary selector gene in L4/L5 that activates different target genes in L4 vs L5 based on the differential availability of open chromatin loci.

Overall, the manuscript presents an interesting example of how Notch activity cooperates with TF expression to generate diverging cell fates. Together with the accompanying paper, it helps thoroughly describe how lamina cell types L4 and L5 are specified and provides an interesting hypothesis for the role of Notch and Bsh in increasing neuronal diversity in the lamina during evolution.

Weaknesses:

Differential Notch activity in L4 and L5:

● The manuscript focuses its attention on describing Notch activity in L4 vs L5 neurons. However, from the data presented, it is very likely that the pool of progenitors (LPCs) is already subdivided into at least two types of progenitors that will rise to L4 and L5, respectively. Evidence to support this is the activity of E(spl)-mɣ-GFP and the Dl puncta observed in the LPC region. Discussion should naturally follow that Notch-induced differences in L4/L5 might preexist L1-expressed Dl that affect newborn L4/L5. Therefore, the differences between L4 and L5 fates might be established earlier than discussed in the paper. The authors should acknowledge this possibility and discuss it in their model.

● The authors claim that Notch activation is caused by L1-expressed Delta. However, they use an LPC driver to knock down Dl. Dl-KD should be performed exclusively in L1, and the fate of L4 should be assessed.

● To test whether L4 neurons are derived from NotchON LPCs, I suggest performing MARCM clones in early pupa with an E(spl)-mɣ-GFP reporter.

● The expression of different Notch targets in LPCs and L4 neurons may be further explored. I suggest using different Notch-activity reporters (i.e., E(spl)-GFP reporters) to further characterize these differences. What cause the switch in Notch target expression from LPCs to L4 neurons should be a topic of discussion.

Notch role in establishing L4 vs L5 fates:

● The authors describe that 27G05-Gal4 causes a partial Notch Gain of Function caused by its genomic location between Notch target genes. However, this is not further elaborated. The use of this driver is especially problematic when performing Notch KD, as many of the resulting neurons express Ap, and therefore have some features of L4 neurons. Therefore, Pdm3+/Ap+ cells should always be counted as intermediate L4/L5 fate (i.e., Fig3 E-J, Fig3-Sup2), irrespective of what the mechanistic explanation for Ap activation might be. It's not accurate to assume their L5 identity. In Fig4 intermediate-fate cells are correctly counted as such.

● Lines 170-173: The temporal requirement for Notch activity in L5-to-L4 transformation is not clearly delineated. In Fig4-figure supplement 1D-E, it is not stated if the shift to 29{degree sign}C is performed as in Fig4-figure supplement 1A-C.

● Additionally, using the same approach, it would be interesting to explore the window of competence for Notch-induced L5-to-L4 transformation: at which point in L5 maturation can fate no longer be changed by Notch GoF?

L4-to-L3 conversion in the absence of Bsh

● Although interesting, the L4-to-L3 conversion in the absence of Bsh is never shown to be dependent on Notch activity. Importantly, L3 NotchON status is assumed based on their position next to Dl-expressing L1, but it is not empirically tested. Perhaps screening Notch target reporter expression in the lamina, as suggested above, could inform this issue.

● Otherwise, the analysis of Bsh Loss of Function in L4 might be better suited to be included in the accompanying manuscript that specifically deals with the role of Bsh as a selector gene for L4 and L5.

Different chromatin landscape in L4 and L5 neurons

● A major concern is that, although L4 and L5 neurons are shown to present different chromatin landscapes (as expected for different neuronal types), it is not demonstrated that this is caused by Notch activity. The paper proves unambiguously that Notch activity, in concert with Bsh, causes the fate choice between L4 and L5. However, that this is caused by Notch creating a differential chromatin landscape is based only in correlation (NotchON cells having a different profile than NotchOFF). Although the authors are careful not to claim that differential chromatin opening is caused directly by Notch, this is heavily suggested throughout the text and must be toned down.

e.g.: Line 294: "With Notch signaling, L4 neurons generate distinct open chromatin landscape" and Line 298: "Our findings propose a model that the unique combination of HDTF and open chromatin landscape (e.g. by Notch signaling)" . These claims are not supported well enough, and alternative hypotheses should be provided in the discussion. An alternative hypothesis could be that LPCs are already specified towards L4 and L5 fates. In this context, different early Bsh targets in each cell type could play a pioneer role generating a differential chromatin landscape.

● The correlation between open chromatin and Bsh loci with Differentially Expressed genes is much higher for L4 than L5. It is not clear why this is the case, and should be discussed further by the authors.

---

## [Author Response]

The following is the authors’ response to the original reviews.

**Public Reviews:**

**Reviewer #1 (Public Review):**
Like the "preceding" co-submitted paper, this is again a very strong and interesting paper in which the authors address a question that is raised by the finding in their co-submitted paper - how does one factor induce two different fates. The authors provide an extremely satisfying answer - only one subset of the cells neighbors a source of signaling cells that trigger that subset to adopt a specific fate. The signal here is Delta and the read-out is Notch, whose intracellular domain, in conjunction with, presumably, SuH cooperates with Bsh to distinguish L4 from L5 fate (L5 is not neighbored by signalproviding cells). Like the back-to-back paper, the data is rigorous, well-presented and presents important conclusions. There's a wealth of data on the different functions of Notch (with and without Bsh). All very satisfying.

Thanks!

I have again one suggestion that the authors may want to consider discussing. I'm wondering whether the open chromatin that the author convincingly measure is the CAUSE or the CONSEQUENCE of Bsh being able to activate L4 target genes. What I mean by this is that currently the authors seem to be focused on a somewhat sequential model where Notch signaling opens chromatin and this then enables Bsh to activate a specific set of target genes. But isn't it equally possible that the combined activity of Bsh/Notch(intra)/SuH opens chromatin? That's not a semantic/minor difference, it's a fundamentally different mechanism, I would think. This mechanism also solves the conundrum of specificity - how does Notch know which genes to "open" up? It would seem more intuitive to me to think that it's working together with Bsh to open up chromatin, with chromatin accessibility than being a "mere" secondary consequence. If I'm not overlooking something fundamental here, there is actually also a way to distinguish between these models - test chromatin accessibility in a Bsh mutant. If the author's model is true, chromatin accessibility should be unchanged.I again finish by commending the authors for this terrific piece of work.

Thanks! It is a crucial question whether Notch signaling regulates chromatin landscape independently of a primary HDTF. We will include this discussion in the text and pursue it in our next project.

We think Notch signaling may regulate chromatin accessibility independently of a primary HDTF based on our observation: in larval ventral nerve cord, all premotor neurons are NotchON neurons while all postsensory neurons are NotchOFF neurons; NotchON neurons share similar functional properties, despite expressing distinct HDTFs, possibly due to the common chromatin landscape regulated by Notch signaling.

**Reviewer #2 (Public Review):**
Summary:In this work, the authors explore how Notch activity acts together with Bsh homeodomain transcription factors to establish L4 and L5 fates in the lamina of the visual system of *Drosophila*. They propose a model in which differential Notch activity generates different chromatin landscapes in presumptive L4 and L5, allowing the differential binding of the primary homeodomain TF Bsh (as described in the cosubmitted paper), which in turn activates downstream genes specific to either neuronal type. The requirement of Notch for L4 vs. L5 fate is well supported, and complete transformation from one cell type into the other is observed when altering Notch activity. However, the role of Notch in creating differential chromatin landscapes is not directly demonstrated. It is only based on correlation, but it remains a plausible and intriguing hypothesis.

Thanks for the positive feedback!

Strengths:The authors are successful in characterizing the role of Notch to distinguish between L4 and L5 cell fates. They show that the Notch pathway is active in L4 but not in L5. They identify L1, the neuron adjacent to L4 as expressing the Delta ligand, therefore being the potential source for Notch activation in L4. Moreover, the manuscript shows molecular and morphological/connectivity transformations from one cell type into the other when Notch activity is manipulated.

Thanks!

Using DamID, the authors characterize the chromatin landscape of L4 and L5 neurons. They show that Bsh occupies distinct loci in each cell type. This supports their model that Bsh acts as a primary selector gene in L4/L5 that activates different target genes in L4 vs L5 based on the differential availability of open chromatin loci.

Thanks!

Overall, the manuscript presents an interesting example of how Notch activity cooperates with TF expression to generate diverging cell fates. Together with the accompanying paper, it helps thoroughly describe how lamina cell types L4 and L5 are specified and provides an interesting hypothesis for the role of Notch and Bsh in increasing neuronal diversity in the lamina during evolution.

Thanks for the positive feedback on both manuscripts.

Weaknesses:Differential Notch activity in L4 and L5:● The manuscript focuses its attention on describing Notch activity in L4 vs L5 neurons. However, from the data presented, it is very likely that the pool of progenitors (LPCs) is already subdivided into at least two types of progenitors that will rise to L4 and L5, respectively. Evidence to support this is the activity of E(spl)-mɣ-GFP and the Dl puncta observed in the LPC region. Discussion should naturally follow that Notch-induced differences in L4/L5 might preexist L1-expressed Dl that affect newborn L4/L5. Therefore, the differences between L4 and L5 fates might be established earlier than discussed in the paper. The authors should acknowledge this possibility and discuss it in their model.

We agree. Historically, LPCs are thought to be homogenous; our data suggests otherwise. We now emphasize this in the Discussion as requested. We are also investigating this question using single-cell RNAseq on LPCs to look for molecular heterogeneities. Nevertheless, whether L4 is generated by E(spl)mɣ-GFP+ (NotchON) LPCs does not affect our conclusion that Notch signaling and the primary HDTF Bsh are integrated to specify L4 fate over L5.

● The authors claim that Notch activation is caused by L1-expressed Delta. However, they use an LPC driver to knock down Dl. Dl-KD should be performed exclusively in L1, and the fate of L4 should be assessed.

Dl is transiently expressed in newborn L1 neurons. To knock down Dl in newborn L1, we need to express Dl-RNAi before the onset of Dl expression in newborn L1; the only known Gal4 line expressed that early is the LPC-Gal4, which is the one that we used.

● To test whether L4 neurons are derived from NotchON LPCs, I suggest performing MARCM clones in early pupa with an E(spl)-mɣ-GFP reporter.

We agree! Whether L4 neurons are derived from NotchON LPCs is a great question. However, MARCM clones in early pupa with an E(spl)-mɣ-GFP reporter will not work because E(spl)-mɣ-GFP reporter is only expressed in LPCs but not lamina neurons. We now mention this in the Discussion.

● The expression of different Notch targets in LPCs and L4 neurons may be further explored. I suggest using different Notch-activity reporters (i.e., E(spl)-GFP reporters) to further characterize these. differences. What cause the switch in Notch target expression from LPCs to L4 neurons should be a topic of discussion.

Thanks! It is a great question why Notch induces Espl-mɣ in LPCs but Hey in newborn neurons. However, it is not the question we are tackling in this paper and it will be a great direction to pursue in future. We will add this to our Discussion.

Notch role in establishing L4 vs L5 fates:● The authors describe that 27G05-Gal4 causes a partial Notch Gain of Function caused by its genomic location between Notch target genes. However, this is not further elaborated. The use of this driver is especially problematic when performing Notch KD, as many of the resulting neurons express Ap, and therefore have some features of L4 neurons. Therefore, Pdm3+/Ap+ cells should always be counted as intermediate L4/L5 fate (i.e., Fig3 E-J, Fig3-Sup2), irrespective of what the mechanistic explanation for Ap activation might be. It's not accurate to assume their L5 identity. In Fig4 intermediate-fate cells are correctly counted as such.

We disagree that the use of 27G05-Gal4 is problematic when performing Notch-KD because our conclusion from Notch-KD is that Bsh without Notch signaling activates Pdm3 and specifies L5 fate. However, 27G05-Gal4 does not have any effect on Pdm3 expression. To make this clearer, we will quantify the percentage of Pdm3+ L5 neurons in Bsh+ lamina neurons for Notch-KD experiment. We are sorry this wasn't clearer.

● Lines 170-173: The temporal requirement for Notch activity in L5-to-L4 transformation is not clearly delineated. In Fig4-figure supplement 1D-E, it is not stated if the shift to 29{degree sign}C is performed as in Fig4-figure supplement 1A-C.

Thank you for catching this. We will correct it in the text.

● Additionally, using the same approach, it would be interesting to explore the window of competence for Notch-induced L5-to-L4 transformation: at which point in L5 maturation can fate no longer be changed by Notch GoF?

Our data show that Bsh with transient Notch signaling in newborn neurons specifies L4 fate while Bsh without Notch signaling in newborn neurons specifies L5 fate. Therefore, we think the window of fate competence is during newborn neurons.

However, as suggested by the reviewer, we did the experiment (see figure below). We used Gal80 (Gal80 inhibits Gal4 activity at 18C) to temporarily control Bsh-Gal4 activity for expressing N-ICD (the active form of Notch) in L5 neurons. We found that tub-Gal80ts, Bsh-Gal4>UAS-N-ICD is unable to induce ectopic L4 neurons when we shift the temperature from 18C to 30C to inactivate Gal80 at 15 hours after pupal formation, which is close to the end of lamina neurogenesis. However, it is unknown how many hours it takes to inactivate Gal80 and activate Bsh-Gal4 and thus we decided not to include this data in our manuscript.

L4-to-L3 conversion in the absence of Bsh● Although interesting, the L4-to-L3 conversion in the absence of Bsh is never shown to be dependent on Notch activity. Importantly, L3 NotchON status is assumed based on their position next to Dlexpressing L1, but it is not empirically tested. Perhaps screening Notch target reporter expression in the lamina, as suggested above, could inform this issue.

Our data show the L4-to-L3 conversion in the absence of Bsh and in the presence of Notch activity while the L5-to-L1 conversion in the absence of Bsh and in the absence of Notch activity. Therefore, Notch activity is necessary for the L4-to-L3 conversion. Unfortunately, currently, we only have Hey as an available Notch target reporter in newborn neurons. To tackle this challenge in the future, we will profile the genome-binding targets of endogenous Notch in newborn neurons. This will identify novel genes as Notch signaling reporters in neurons for the field.

● Otherwise, the analysis of Bsh Loss of Function in L4 might be better suited to be included in the accompanying manuscript that specifically deals with the role of Bsh as a selector gene for L4 and L5.

That is an interesting suggestion, but without knowing that Bsh + Notch = L4 identity the experiment would be hard to interpret. Note that we took advantage of Notch signaling to trace the cell fate in the absence of Bsh and found the L4-to-L3 conversion (see Figure 5G-K).

Different chromatin landscape in L4 and L5 neurons● A major concern is that, although L4 and L5 neurons are shown to present different chromatin landscapes (as expected for different neuronal types), it is not demonstrated that this is caused by Notch activity. The paper proves unambiguously that Notch activity, in concert with Bsh, causes the fate choice between L4 and L5. However, that this is caused by Notch creating a differential chromatin landscape is based only in correlation. (NotchON cells having a different profile than NotchOFF). Although the authors are careful not to claim that differential chromatin opening is caused directly by Notch, this is heavily suggested throughout the text and must be toned down.e.g.: Line 294: "With Notch signaling, L4 neurons generate distinct open chromatin landscape" and Line 298: "Our findings propose a model that the unique combination of HDTF and open chromatin landscape (e.g. by Notch signaling)" . These claims are not supported well enough, and alternative hypotheses should be provided in the discussion. An alternative hypothesis could be that LPCs are already specified towards L4 and L5 fates. In this context, different early Bsh targets in each cell type could play a pioneer role generating a differential chromatin landscape.

We agree and appreciate the comment, it is well justified. We have toned down our comments and clearly state that this is a correlation that needs to be tested for a causal relationship. The reviewer posits: “An alternative hypothesis: different early Bsh targets in each cell type could play a pioneer role generating a differential chromatin landscape.” Yes, it is a crucial question whether Notch signaling regulates chromatin landscape independently of a primary HDTF (e.g., Bsh). We will include this discussion in the text and pursue it in our next project. We think Notch signaling may regulate chromatin accessibility independently of a primary HDTF based on our observation: in larval ventral nerve cord, all premotor neurons are NotchON neurons while all post-sensory neurons are NotchOFF neurons; NotchON neurons share similar functional properties, despite expressing distinct HDTFs, possibly due to the common chromatin landscape regulated by Notch signaling.

● The correlation between open chromatin and Bsh loci with Differentially Expressed genes is much higher for L4 than L5. It is not clear why this is the case, and should be discussed further by the authors.

We agree and think in L5 neurons, the secondary HDTF Pdm3 also contributes to L5-specific gene transcription during the synaptogenesis window, in addition to Bsh. We will include this in the text.